# Generation of a CRISPR activation mouse that enables modelling of aggressive lymphoma and interrogation of venetoclax resistance

Yexuan Deng[1,2,6], Sarah T. Diepstraten[2,3,6], Margaret A. Potts [2,3], Göknur Giner[2,3], Stephanie Trezise [2,3], Ashley P. Ng [2,3], Gerry Healey[2,3], Serena R. Kane[2,3], Amali Cooray [2,3], Kira Behrens [2,3], Amy Heidersbach[4], Andrew J. Kueh[2,3], Martin Pal [2,3,5], Stephen Wilcox[2,3], Lin Tai[2], Warren S. Alexander[2,3], Jane E. Visvader [2,3], Stephen L. Nutt [2,3], Andreas Strasser [2,3], Benjamin Haley[4], Quan Zhao [1,7], Gemma L. Kelly [2,3,7] & Marco J. Herold [2,3,7] ✉

CRISPR technologies have advanced cancer modelling in mice, but CRISPR activation (CRISPRa) methods have not been exploited in this context. We establish a CRISPRa mouse (*dCas9a-SAM^{KI}*) for inducing gene expression in vivo and in vitro. Using *dCas9a-SAM^{KI}* primary lymphocytes, we induce B cell restricted genes in T cells and vice versa, demonstrating the power of this system. There are limited models of aggressive double hit lymphoma. Therefore, we transactivate pro-survival BCL-2 in *Eμ-Myc^{T/+};dCas9a-SAM^{KI/+}* haematopoietic stem and progenitor cells. Mice transplanted with these cells rapidly develop lymphomas expressing high BCL-2 and MYC. Unlike standard *Eμ-Myc* lymphomas, BCL-2 expressing lymphomas are highly sensitive to the BCL-2 inhibitor venetoclax. We perform genome-wide activation screens in these lymphoma cells and find a dominant role for the BCL-2 protein A1 in venetoclax resistance. Here we show the potential of our CRISPRa model for mimicking disease and providing insights into resistance mechanisms towards targeted therapies.

The CRISPR/Cas9 system is an elegant tool for genome engineering. While initial usage was restricted to mutation induced loss-of-function applications, recent advances have facilitated more sophisticated forms of genetic manipulation. One example is the CRISPR activation (CRISPRa) system which utilises a Cas9 variant lacking its enzymatic activity (dCas9), thereby converting Cas9 to an inert DNA-binding protein scaffold[1]. Fusion of the VP64[2] activation domain to dCas9 produces a transcriptional activator that can be targeted to any promoter of any gene through a sequence specific single guide RNA (sgRNA) in the presence of a protospacer adjacent motive (PAM, NGG)

[1]The State Key Laboratory of Pharmaceutical Biotechnology, Department of Hepatobiliary Surgery, The Affiliated Drum Tower Hospital of Nanjing University Medical School, School of Life Sciences, Nanjing University, Nanjing, Jiangsu, China. [2]The Walter and Eliza Hall Institute of Medical Research, Melbourne, VIC, Australia. [3]Department of Medical Biology, University of Melbourne, Melbourne, VIC, Australia. [4]Department of Molecular Biology, Genentech, Inc., South San Francisco, CA, USA. [5]School of Dentistry and Medical Sciences, Charles Sturt University, Wagga Wagga, NSW, Australia. [6]These authors contributed equally: Yexuan Deng, Sarah T. Diepstraten. [7]These authors jointly supervised this work: Quan Zhao, Gemma L. Kelly, Marco J. Herold. ✉e-mail: herold@wehi.edu.au

located downstream of the sgRNA binding sequence[3–5]. Early reports showed promising results for gene activation using dCas9-VP64 at some genetic loci, but low to no gene induction at others[3–6]. This inconsistency across different genomic loci sparked the development of more potent dCas9 transactivators[7–10]. One such effective CRISPRa system is the Synergistic Activation Mediator (SAM). It relies on the dCas9-VP64 fusion protein and two additional transcriptional activator domains, p65 and HSF1, that can be recruited into the complex by the MS2 RNA binding protein through loops in the sgRNA scaffold, leading to strong induction of expression of targeted genes[9].

An exciting application of CRISPRa is the potential to model complex human diseases[11]. Cancer is a highly heterogenous disease with many different signalling pathways deregulated within the same cancer type[12]. The ability to create pre-clinical disease models that faithfully reflect the hallmarks of human disease is critical for the identification of specific cancer drivers and/or therapy resistance factors[13]. However, mouse models that accurately mimic aggressive lymphomas, such as double hit lymphoma (DHL), an aggressive subset (~10%) of diffuse large B cell lymphomas (DLBCLs) that express high levels of both c-MYC and BCL-2 due to chromosomal translocations, are lacking[14,15]. Previous attempts to model this disease were limited to humanised or human xenograft models[16–18], which are less amenable to genetic modifications and require an immune compromised environment. Other approaches using *Eμ-Myc/Eμ-Bcl-2* double transgenic mice[19] or ectopic expression of BCL-2 in a *Eμ-Myc* transgenic background[20] failed to recapitulate DHL, the former instead giving rise to lymphoid progenitor tumours.

Here we demonstrate the utility of a CRISPRa mouse model and its applicability in primary cells of the haematopoietic system. Initially, we show gene induction in primary B and T cells derived from the heterozygous and homozygous *dCas9a-SAM^KI* mice transduced with sgRNAs targeting genes such as CD19 and CD4. While these genes can be further induced in their respective lineages, we are also able to express B cell specific genes in the T cell lineage and vice versa. To test the suitability of our CRISPRa mouse for generating disease models, we induce MDM2 expression (the natural antagonist of the tumour suppressor TRP53) in haematopoietic stem and progenitor cells (HSPCs) from CRISPRa enabled *Eμ-Myc* transgenic mice. Following transplantation of these cells into lethally irradiated mice, accelerated tumour onset is observed compared to mice transplanted with control sgRNA transduced HSPCs from CRISPRa enabled *Eμ-Myc* transgenic mice. Next we set out to develop an aggressive lymphoma model, similar to DHL. This is achieved through activation of endogenous BCL-2 expression in *Eμ-Myc/dCas9a-SAM^KI/+* HSPCs followed by their transplantation into lethally irradiated mice. Rapid and aggressive lymphomas develop that are characterised by CD19/B220 double positive tumour cells and the expression of high levels of both c-MYC and BCL-2, which are all markers of human DHL[21]. These lymphomas are highly sensitive to treatment with venetoclax, the BH3-mimetic drug that specifically binds and inhibits pro-survival BCL-2. Venetoclax is FDA approved for the therapy of Chronic Lymphocytic Leukemia (CLL) and Acute Myeloid Leukemia (AML) and is in clinical trials for several other cancers[22]. Identifying resistance factors to venetoclax therapy is therefore an area of great clinical relevance. Whilst CRISPR gene knock-out screens have identified TRP53 loss and other factors that are mediators of venetoclax resistance[23,24], no studies have utilised CRISPRa to investigate genes that can confer venetoclax resistance when upregulated from their endogenous promoters. Cell lines derived from our CRISPRa mouse model of aggressive DHL-like lymphomas that are highly sensitive to venetoclax provide an ideal platform to perform genome-wide CRISPRa screens for identifying resistance factors. These screens reveal a dominant role for the relatively poorly studied pro-survival protein A1 (called BFL-1 in humans) in conferring potent resistance to venetoclax.

## Results

### Establishment of a robust CRISPR activation platform

CRISPRa has the potential to be used for the development of accurate disease models because the induction of oncogenic driver genes from their endogenous promoters is physiologically more relevant compared to overexpression of a gene with cDNA constructs. We therefore sought to establish a robust and widely applicable CRISPRa system for the generation of faithful pre-clinical cancer models and the identification of targets that could be translated into improved therapies for cancer patients. We adapted the SAM system originally described in a two-vector configuration to be expressed from a single construct to achieve similar expression of all the components required for CRISPR mediated gene induction[9]. To this end, we linked the dCas9-VP64 via a T2A sequence to the MS2-p65-HSF1 expression cassette (Fig. 1a). In addition, we incorporated an eGFP sequence as a marker via a second T2A sequence downstream of the dCas9-VP64-T2A-MS2-p65-HSF1 coding sequence. We incorporated this cassette into a lentiviral vector for ease of manipulating cells[25]. To validate the efficiency of this CRISPRa cassette in vitro, we introduced lentiviral vectors encoding the CRISPRa cassette and one of three unique sgRNAs targeting the *Bcl-2* promoter into cell lines derived from the *Eμ-Myc* transgenic mouse model of lymphoma[26] (Supplementary Fig. 1a). Western blot analysis of two independent lymphoma cell lines confirmed that all sgRNAs caused a substantial increase in BCL-2 expression, detectable even before puromycin selection of sgRNA-transduced cells (Fig. 1b and Supplementary Fig. 1b).

*Eμ-Myc* lymphomas are highly reliant on the pro-survival protein MCL-1 for their sustained survival[27–29]. The elevated BCL-2 expression observed in the CRISPRa transduced lymphoma cells substituted for MCL-1 and hence increased resistance to the MCL-1 selective inhibitor S63845 (Fig. 1c). Interestingly, the enforced expression of BCL-2 in these lymphoma cells did not sensitise them to the BCL-2 inhibitor venetoclax (Fig. 1d), likely because these tumour cells did not develop under conditions of high BCL-2 expression and therefore do not depend on BCL-2 expression for continued survival. These experiments confirmed that our CRISPRa cassette could induce strong upregulation of targeted genes in cell lines in vitro but also highlighted the limitations of using established cell lines for studies relevant to more sophisticated cancer models. We therefore sought to progress this system to an in vivo setting, by generating transgenic mice with a similar configuration for CRISPRa mediated transcriptional upregulation.

### Generation of CRISPRa transgenic mice

For flexible expression of the CRISPRa system in vivo, we targeted the dCas9-VP64-T2A-MS2-p65-HSF1 cassette into the ubiquitously expressed *Rosa26* locus. We used the CTV vector[30], which has a loxP flanked stop cassette between the CAG promoter and the cDNA, allowing for temporal or cell type specific induction of gene expression upon CRE mediated deletion (Fig. 2a). Crossing the CRISPRa transgenic mice to a CRE deleter mouse strain removed the stop cassette, allowing expression of the CRISPRa components in all tissues. We refer to the resulting strain as *dCas9a-SAM*. Transgene insertion and expression were confirmed by long range PCR (LR-PCR) and Southern Blotting on DNA isolated from *dCas9a-SAM* transgenic mice, and flow cytometric analysis for eGFP on cells isolated from the thymus, spleen, bone marrow and lymph nodes (Fig. 2b, c and Supplementary Fig. 2). Moreover, intracellular staining and flow cytometric analysis of thymocytes and bone marrow cells, and Western blot analysis of liver, kidney and heart with Cas9-specific antibodies, confirmed that the CRISPRa components were expressed in diverse tissue types (Fig. 2d and Supplementary Fig. 3a–c).

Since the dCas9 protein is fused to the VP64 activation domain, we next sought to determine whether the constitutive expression of a transcriptional activator protein had any impact on the cellular

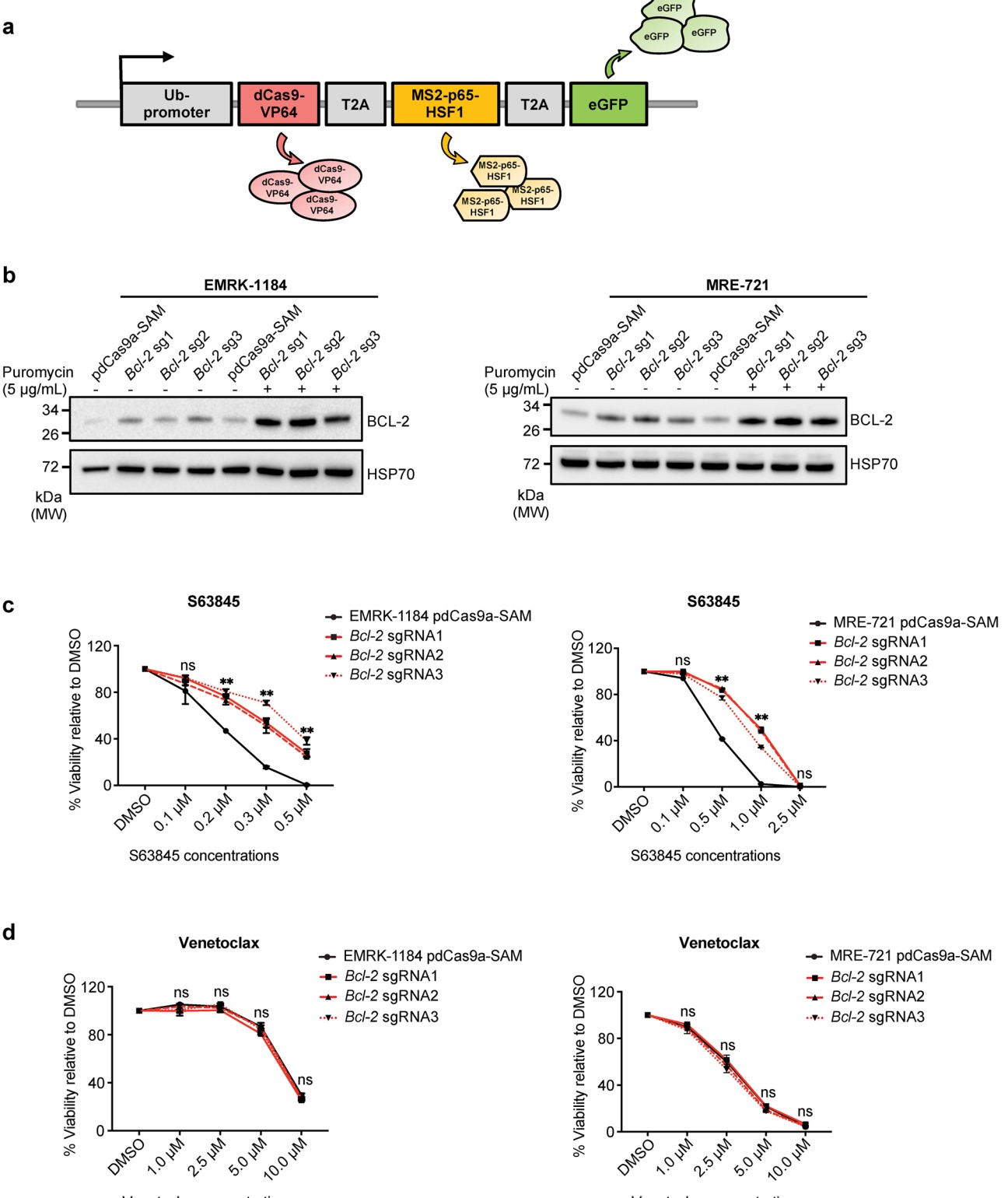

**Fig. 1 | Generation and validation of pdCas9a-SAM construct in *E*μ-*Myc* lymphoma cell lines. a** Schematic representation of the pdCas9a-SAM lentiviral construct. **b** Western blot analysis for BCL-2 in manipulated *E*μ-*Myc* lymphoma cell lines. EMRK-1184 or MRE-721 *E*μ-*Myc* lymphoma-derived cell lines were transduced with pdCas9a-SAM only or pdCas9a-SAM plus *Bcl-2* sgRNAs. Cell lysates were harvested from transduced cell lines, before or after puromycin-selection and expression of the indicated proteins was examined by Western blotting. Probing for the heat shock protein 70 (HSP70) served as a loading control. 2 independent experiments were performed showing similar results. **c, d** Viability of *E*μ-*Myc*

lymphoma cell lines transduced with pdCas9a-SAM only or pdCas9a-SAM plus *Bcl-2* sgRNAs. Cells were treated with the MCL-1 inhibitor S63845 or the BCL-2 inhibitor venetoclax at the indicated drug concentrations. Cell viability was determined at 24 h by propidium iodide (PI) staining and subsequent flow cytometric analysis. Data are represented as mean ± SD, $n = 3$ independent experiments. The statistical significance was determined using two-way ANOVA test with EMRK-1184 pdCas9a-SAM or MRE-721 pdCas9a-SAM cells serving as the control. Exact *P* values are provided in the Source Data file. Overall *P* values are shown as ns = no significant difference, **$P < 0.01$. Source data are provided as a Source Data file.

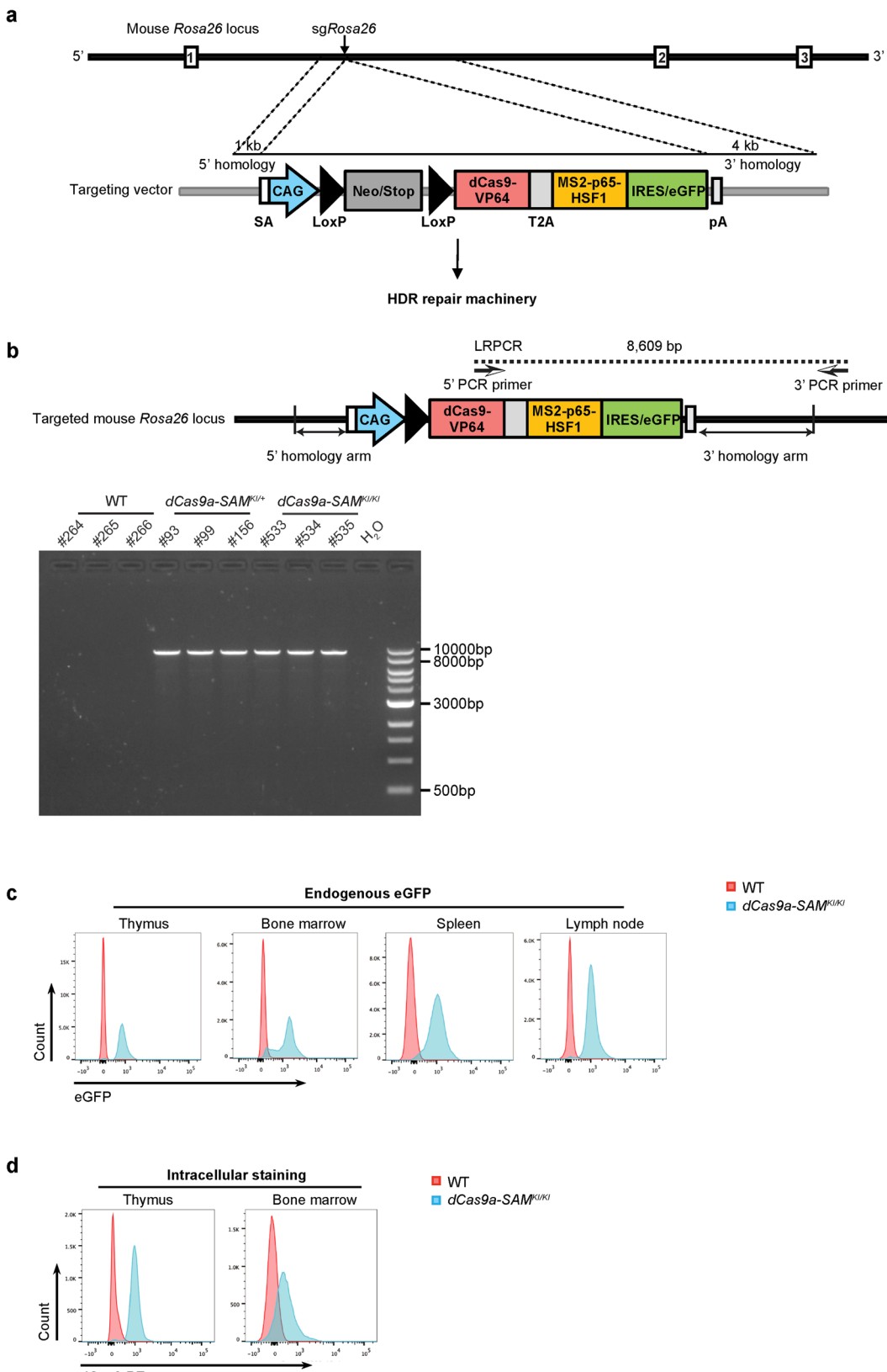

**Fig. 2 | Establishment and validation of *dCas9a-SAM^(KI/KI)* mice. a** Strategy for targeting of the dCas9a-SAM cassette into the mouse *Rosa26* locus using CRISPR/Cas9 technology. Repair of the DNA cut by endogenous homology-directed repair (HDR) mechanism using homology sequences present in the pRosa26-dCas9a-SAM vector enables the insertion of the dCas9a-SAM cassette into the *Rosa26* locus. **b** Long range PCR (LR–PCR) based validation of dCas9 expression in *dCas9a-SAM* transgenic mice. The expected product is 8609 bp. 2 independent experiments were performed showing similar results. **c, d** Representative flow cytometry data on eGFP expression or intracellular dCas9 staining in the indicated haematopoietic tissues derived from wildtype (WT, negative control) or *dCas9a-SAM^(KI/KI)* mice. 3 WT mice and 4 *dCas9a-SAM^(KI/KI)* mice were used for the analysis showing similar results. Source data are provided as a Source Data file.

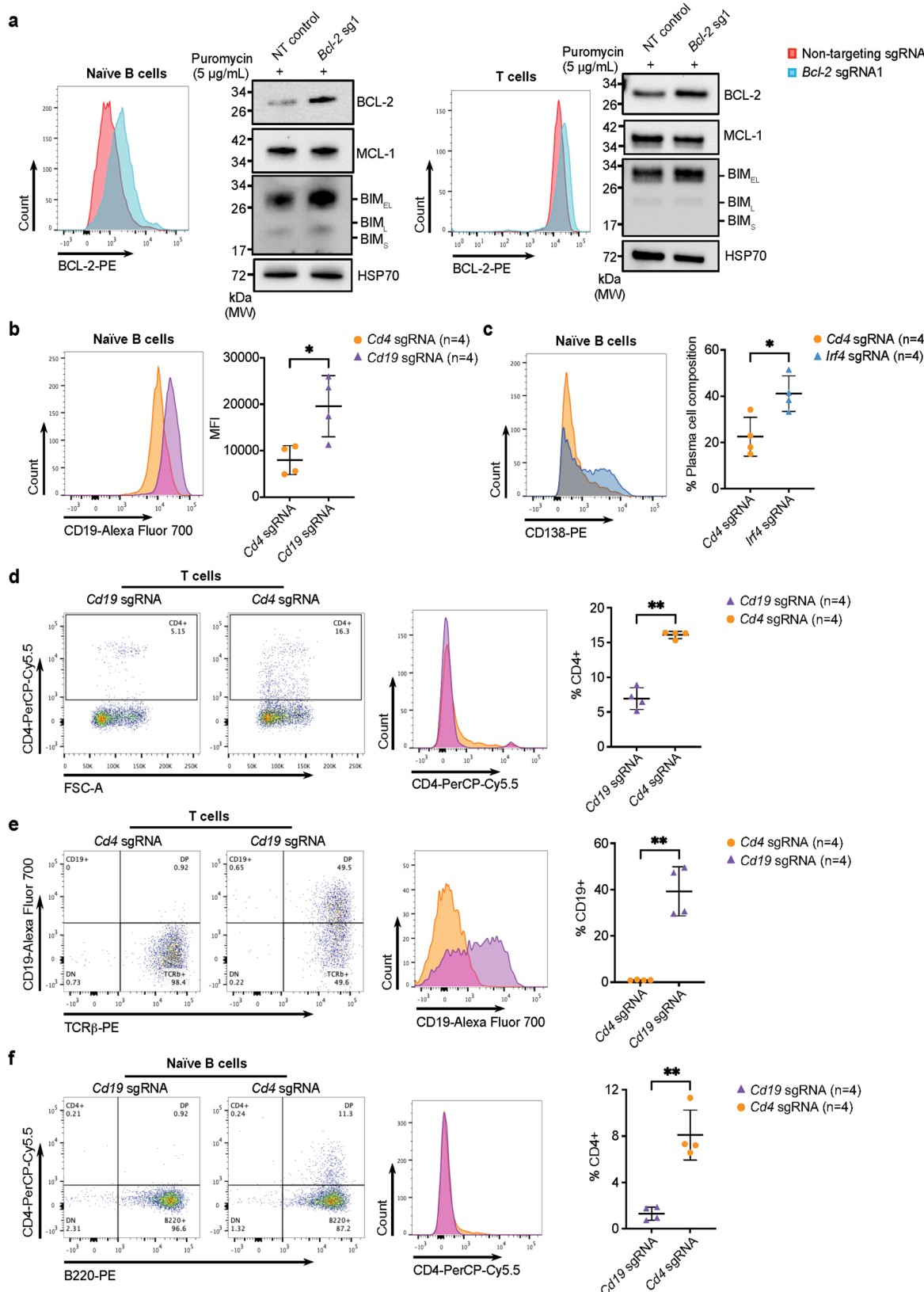

composition of the mice. We analysed the haematopoietic compartment because it is particularly sensitive to cytotoxic stress and even subtle changes in gene expression. We found that homozygous *dCas9a-SAM*[KI/KI] mice showed no differences in numbers of the different immune cell subsets in the thymus, bone marrow, spleen and lymph nodes compared to wildtype mice (Supplementary Fig. 4).

Importantly, aged (older than 12 months) *dCas9a-SAM*[KI/KI] mice showed no signs of disease, further confirming that constitutive expression of the CRISPRa system in all tissues of the animals does not cause marked toxicity or substantive changes to the proportions of all tested cell types. Finally, we performed RNA-seq analysis on wildtype (WT) and *dCas9a-SAM*[KI/+] mouse embryonic fibroblasts (MEFs) transduced with a

**Fig. 3 | Induction of robust gene expression in primary murine B cells and T cells.** Naïve B cells or splenocytes were isolated from the *dCas9a-SAM^KI/KI^* mice and cultured in vitro in medium containing LPS or ConA plus IL2, respectively. **a** Mitogen activated B or T cells were transduced with the *Bcl-2* sgRNA1 and selected with 5 µg/mL puromycin for 2 days. Robust BCL-2 expression was validated by intracellular staining for BCL-2 protein by FACS and by Western blot analysis. Probing for HSP70 served as a loading control on Western blots. 3 independent experiments were performed showing similar results. **b**, **c** Activated B cells were transduced with *Cd19* or *Irf4* sgRNAs. The expression of CD19 or CD138 (impacted by IRF4 expression) was analysed by flow cytometry. Cells transduced with *Cd4* sgRNAs served as a negative control. Data are presented as mean ± SD, n = 4 independent experiments. The statistical significance was determined by two-sided student's *t*-test. P = 0.0189 (**b**), P = 0.0170 (**c**), *P < 0.05. **d**, **e** Activated T cells were transduced with *Cd4* or *Cd19* sgRNAs. T cells were gated on TCR β⁺. The expression of CD4 or CD19 was analysed by flow cytometry. Cells transduced with *Cd19* sgRNAs served as negative controls for CD4 upregulation and vice versa. Data are presented as mean ± SD, n = 4 independent experiments. The statistical significance was determined by two-sided student's *t*-test. P < 0.0001 (**d**), P = 0.0004 (**e**), **P < 0.01. **f** Activated B cells were transduced with *Cd4* sgRNAs. B cells were gated on B220⁺. The expression of CD4 was analysed by flow cytometry. Cells transduced with *Cd19* sgRNAs served as the negative control. Data are presented as mean ± SD, n = 4 independent experiments. The statistical significance was determined by two-sided student's *t*-test. P = 0.0009, **P < 0.01. Source data are provided as a Source Data file.

non-targeting control sgRNA (sgNT) to verify that our CRISPRa system does not inherently induce off-target gene activation (Supplementary Fig. 3d). Importantly, this analysis showed no major differences in the transcriptional expression profile between the WT and *dCas9a-SAM^KI/+^/sgNT* expressing cells.

Next, we assessed the efficiency of the CRISPRa components for the activation of targeted gene expression in primary cells derived from the *dCas9a-SAM* mice. To this end, we stimulated purified naïve B cells or splenocytes isolated from *dCas9a-SAM^KI/KI^* mice with LPS or Concanavalin A plus Interleukin-2 for the generation of activated B or T cell blasts, respectively. We then transduced these cells with the *Bcl-2* sgRNA1 construct (that co-expresses puromycin resistance) and selected transduced cells with puromycin for 2 days. Analysis of BCL-2 expression by intracellular flow cytometry and Western blotting revealed upregulation of BCL-2 in both activated B and T cells (Fig. 3a).

To assess the potential of the CRISPRa system for targeting various genes, we introduced sgRNAs for CD19 or IRF4 into B cells. As expected, CD19-specific sgRNAs elevated surface expression of CD19 above the basal level, while introducing sgRNAs for IRF4, known to induce differentiation of B cells into plasma cells[31], enhanced the frequency of cells positive for the plasma cell marker CD138 (Fig. 3b, c). Similarly, transducing activated T cells with sgRNAs targeting the CD4 promoter enhanced expression of CD4 on the cell surface (Fig. 3d). Since these genes are already transcriptionally active in B cells or T cells, we next challenged our CRISPRa system by attempting to induce expression of genes that are normally transcriptionally silent in these cells. We first introduced sgRNAs targeting the B cell marker CD19 into T cells. Remarkably, this elicited CD19 expression in almost 50% of T cells (Fig. 3e). Similarly, introduction of sgRNAs targeting the T cell specific CD4 gene promoter into B cells resulted in B cells with CD4 expression (Fig. 3f). In addition, we were able to induce gene expression in cells derived from heterozygous *dCas9a-SAM^KI/+^* mice, i.e. CD19 in T cells, CD4 in B cells and BCL-2 in MEFs (Supplementary Fig. 5a–c). This confirms that a single copy of the CRISPRa system is able to achieve activation of these normally silenced genes and that we can induce gene expression in diverse tissue types. These data demonstrate that we have developed a CRISPRa mouse model that displays no detectable toxicity yet has the potency required to induce the expression of targeted genes in primary cells, even of genes that are normally silenced within a specific cell type.

## Exploiting CRISPRa in vivo for the development of aggressive lymphomas

Having validated the efficiency of the CRISPRa mouse for gene induction in primary cells, we set out to test its applicability for developing disease models. Initially, we sought to confirm whether induction of a gene product by CRISPRa could indeed affect disease aetiology, for example, by modulating the latency of tumour development. We know from previous reports that deleting the *Trp53* tumour suppressor gene dramatically accelerates *Eμ-Myc*-driven tumourigenesis[32]. To replicate the effect of loss of TRP53 activity using CRISPRa, we transduced *Eμ-Myc/dCas9a-SAM^KI/+^* HSPCs with sgRNAs to

induce expression of MDM2, the E3 ligase that targets TRP53 for proteasomal degradation (sg*Mdm2*), or non-targeting sgRNAs as controls. The transduced HSPCs were then transplanted into lethally irradiated C57BL/6-Ly5.1 recipient mice which were monitored for tumour development (Fig. 4a and Supplementary Fig. 6a). As occurs for loss of *Trp53* in the *Eμ-Myc* background, we observed accelerated tumour onset in the mice reconstituted with *Eμ-Myc/dCas9a-SAM^KI/+^/sgMdm2* HSPCs compared to control mice. To determine the levels of TRP53 in the lymphomas expressing sg*Mdm2*, we derived cell lines and induced expression of TRP53 with the MDM2 inhibitor Nutlin3a[33]. Western blot analysis clearly demonstrated a reduction in TRP53 protein levels in *Eμ-Myc/dCas9a-SAM^KI/+^/sgMdm2* lymphomas compared to control lymphoma cells, consistent with elevated MDM2 levels (Supplementary Fig. 6b).

Having validated our CRISPRa system, we next sought to develop an aggressive model of lymphoma. We chose DHL, for which previous attempts to mimic this devastating disease with a mouse model have not been very successful. Such a lymphoma model would facilitate the identification and validation of therapeutic strategies for patients with DHL. To this end, we used the same approach as described above, this time using one of the validated sg*Bcl-2* constructs described in Fig. 1. Irradiated mice were injected with *Eμ-Myc/dCas9a-SAM^KI/+^/sgBcl-2* HSPCs or control *Eμ-Myc/dCas9a-SAM^KI/+^/sgNT* HSPCs, and 6 weeks post-transplantation, BCL-2 expression was analysed in haematopoietic cells of recipient mice by flow cytometry. The analysis revealed increased BCL-2 expression in peripheral blood cells of mice transplanted with *Eμ-Myc/dCas9a-SAM^KI/+^/sgBcl-2* HSPCs (Fig. 4b). Accordingly, these mice went on to develop aggressive lymphomas with a median latency of only 68 days, compared with a median latency of 132 days for control mice transplanted with *Eμ-Myc/dCas9a-SAM^KI/+^/sgNT* HSPCs (Fig. 4c). Characterisation of *Eμ-Myc/dCas9a-SAM^KI/+^/sgBcl-2* lymphomas revealed a B cell phenotype (CD19/B220 double positive; Fig. 4d and Supplementary Fig. 7a) and high BCL-2 expression (Fig. 4e), which are both also observed in human DHL[21]. Additional staining of primary tumour samples for mutations in the tumour suppressor TRP53 showed no marked differences in frequency of tumours with TRP53 mutations between *Eμ-Myc/dCas9a-SAM^KI/+^/sgBcl-2* and control lymphomas (Supplementary Fig. 7b), and is in concordance with human DHL studies in which TP53 mutations have also been reported[34]. Pre-leukaemic analysis of mice transplanted with *Eμ-Myc/dCas9a-SAM^KI/+^/sgBcl-2* HSPCs revealed an overall increase in the number and frequency of B cells, predominantly naïve B cells, compared with mice which received control HSPCs (Supplementary Fig. 7c, d). These *Eμ-Myc/dCas9a-SAM^KI/+^/sgBcl-2* B cell lymphomas could readily be derived into cell lines in vitro (Fig. 5). This model of aggressive lymphoma therefore contrasts with a previously described mouse model that utilised *Eμ-Myc/Eμ-Bcl-2* double transgenic mice which developed lymphomas exhibiting an immature haematopoietic progenitor phenotype rather than a B cell derived tumour[19] and could not be cultured in vitro[35].

For further characterisation and experimentation, cell lines were derived from the *Eμ-Myc/dCas9a-SAM^KI/+^/sgBcl-2* lymphomas and

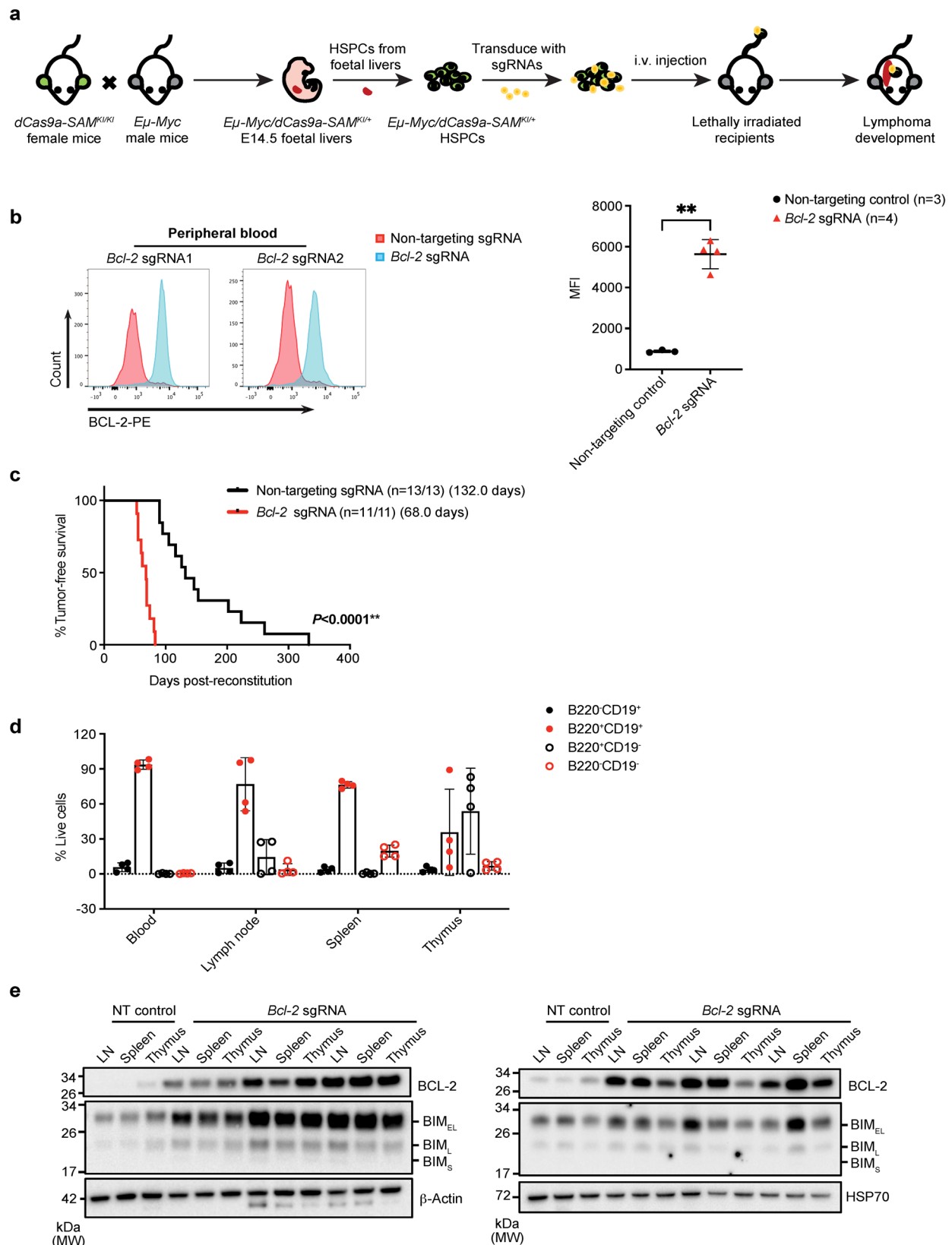

control lymphomas. Interestingly, comparing the BCL-2 levels of *Eμ-Myc/Eμ-Bcl-2* lymphomas (note this is human BCL-2) with our *Eμ-Myc/dCas9a-SAM^{KI/+}/sgBcl-2* lymphomas revealed no detectable differences in the primary tissues, but lower overall expression of BCL-2 protein was observed in the cell lines derived from our lymphomas compared to the primary tumour cells (Supplementary Fig. 8). All sg*Bcl-2*

lymphoma-derived cell lines displayed higher expression of BCL-2 and the pro-apoptotic BH3-only protein BIM, compared with the control cell lines derived from lymphomas containing non-targeting control sgRNAs (#219, #220), as detected by both Western blotting (Fig. 5a) and intracellular flow cytometric analysis (Fig. 5b). We further noted that expression of the related pro-survival protein BCL-XL was

**Fig. 4 | Haematopoietic reconstitution with *E*μ-*Myc*/*dCas9a-SAM*^KI/+^ HSPCs.**
**a** Flow diagram of the haematopoietic reconstitution assay. Female *dCas9a-SAM*^KI/KI^ mice were crossed with male *E*μ-*Myc*^T/+^ mice to produce *E*μ-*Myc*/*dCas9a-SAM*^KI/+^ embryos. HSPCs were isolated from the foetal livers of E14.5 embryos and cultured in vitro for sgRNA transduction. HSPCs were transduced with sgRNAs (*sgMdm2*, *sgBcl-2* or non-targeting control sgRNAs) and then transplanted into lethally irradiated recipient mice by intravenous (i.v.) injection. Enlarged spleen, lymph nodes or thymus, and a high level of white blood cells were considered as signs of malignant disease to identify lymphoma-bearing mice. **b** The expression of BCL-2 in cells from the peripheral blood of reconstituted mice was determined by intracellular staining and flow cytometry. Data are presented as mean ± SD, 3 control mice and 4 *sgBcl-2* reconstituted mice were used for the analysis. The statistical significance was determined by two-sided student's *t*-test. *P* < 0.0001, **P* < 0.01. **c** Kaplan−Meier survival curve of reconstituted mice transplanted with *E*μ-*Myc*/

*dCas9a-SAM*^KI/+^/*sgBcl-2* HSPCs or control HSPCs. n/n = numbers of sick mice/numbers of total recipient mice. Data are combined from two independent reconstitution experiments. The statistical significance was determined by the Mantel-Cox test. *P* < 0.0001, **P* < 0.01. **d** Characterisation of *E*μ-*Myc*/*dCas9a-SAM*^KI/+^/*sgBcl-2* lymphoma phenotype. Blood, lymph node, spleen and thymus were harvested from sick mice for B220/CD19 staining and flow cytometric analysis. Data are presented as mean ± SD, *n* = 4 individual *sgBcl-2* reconstituted mice. **e** The expression of BCL-2 and BIM in tumour tissues of mice transplanted with *E*μ-*Myc*/*dCas9a-SAM*^KI/+^/*sgBcl-2* HSPCs or control HSPCs is shown by Western blotting. Probing for β-actin or HSP70 served as protein loading controls. Lymph node (LN), spleen and thymus tissues from 2 control HSPC reconstituted mice and 6 *sgBcl-2* HSPC reconstituted mice were used for Western blot analysis. 2 independent experiments were performed showing similar results. Source data are provided as a Source Data file.

highly variable across the individual lymphomas (Fig. 5a). More detailed phenotyping of the *E*μ-*Myc*/*dCas9a-SAM*^KI/+^/*sgBcl-2* lymphoma cells showed that they were mostly positive for IgM and CD43 (but negative for IgD), suggesting an immature B cell phenotype (Supplementary Fig. 9). Analysis of primary *E*μ-*Myc*/*dCas9a-SAM*^KI/+^/*sgBcl-2* lymphomas and cell lines generated from these primary tissues revealed a monoclonal origin of malignant cells (Fig. 5c). Significantly, we found that all cell lines derived from *sgBcl-2* lymphomas were sensitive to venetoclax treatment (mean IC50 = 0.11 μM), which is in striking contrast to the resistance to venetoclax (mean IC50 > 1 μM) observed in lymphomas that arise in *E*μ-*Myc* transgenic mice (Fig. 5d and Supplementary Fig. 10 a, b and Supplementary Table 1). In addition, whilst the BCL-2 over-expressing lymphoma lines were less sensitive overall to treatment with the MCL-1 inhibitor S63845 than control *E*μ-*Myc* lymphoma lines, some of the BCL-2 expressing lines that displayed lower venetoclax sensitivity were still similarly sensitive to the MCL-1 inhibitor as control *E*μ-*Myc* lymphomas (Fig. 5e and Supplementary Fig. 10c, d and Supplementary Table 1).

To further characterise our *E*μ-*Myc*/*dCas9a-SAM*^KI/+^/*sgBcl-2* lymphoma cells, we compared the transcriptional profile of our developed lymphomas with that of a murine preB-ALL model[36]. Interestingly, this revealed clear differences in the top 200 most variable genes (Fig. 6a) and provided evidence that our model is similar to DHL/DLBCL as evidenced by upregulation of pathways (e.g. DLBCL, MYC) and specific genes (e.g. *Ep300, Stat6*) that are observed in human DHL (Fig. 6b, Supplementary Table 2 and Supplementary Data 1)[37].

These data confirm that we have established a model of aggressive B cell lymphoma, that has characteristics of human DHL, i.e. monoclonal, double high expression of c-MYC and BCL-2, cell surface expression of CD19, and a transcriptional profile distinct from other B cell tumours and consistent with aggressive human DHLs[21,37]. Our results suggest that both the BCL-2 inhibitor venetoclax and MCL-1 inhibitors (already in clinical trials for B cell malignancies but not DHL)[38] have potential for the treatment of this disease in humans, either as single agents or in combination.

**Using CRISPR activation screens to find venetoclax resistance factors**

An important clinical issue is the emergence of resistance to venetoclax in patients undergoing therapy[22]. The generation of *E*μ-*Myc*/*dCas9a-SAM*^KI/+^/*sgBcl-2* lymphoma cell lines, that are highly sensitive to venetoclax and CRISPRa enabled, provided a model system in which whole genome CRISPR activation screens could be carried out to identify genes that confer drug resistance when upregulated. We transduced six replicates each of two venetoclax sensitive *E*μ-*Myc*/*dCas9a-SAM*^KI/+^/*sgBcl-2* lymphoma lines with a recently described mouse genome-wide CRISPRa sgRNA library[39]. The cells were cultured for two weeks after transduction to permit induction of gene expression and were then subjected to treatment with vehicle (DMSO) or the indicated concentrations of venetoclax for a further two weeks

(Fig. 7a). DNA samples were collected, and next generation sequencing was performed to identify the sgRNAs enriched in venetoclax-treated versus control cell populations. At all concentrations, venetoclax treatment led to a strong enrichment of a subset of sgRNAs compared to the DMSO treated control samples (Supplementary Fig. 11a). Notably, we found enrichment of sgRNAs upregulating two pro-survival BCL-2 family members, BCL-XL and MCL-1, that, based on current literature, would be expected to mediate resistance to venetoclax[22] (Supplementary Fig. 11c). To our surprise, however, we found that sgRNAs targeting the underappreciated pro-survival BCL-2 family member A1 were the most dominant sgRNAs enriched by venetoclax treatment in both cell lines (Fig. 7b and Supplementary Fig. 11b). This was particularly evident at IC80 doses of venetoclax, where multiple sgRNAs targeting A1 were highly significantly enriched (FDR < 0.05) compared to DMSO treated control groups (Supplementary Data 2 and 3). To confirm that upregulation of A1 can confer protection from venetoclax induced killing, sgRNAs targeting the *Bcl2a1* promoter were transduced into two *E*μ-*Myc*/*dCas9a-SAM*^KI/+^/*sgBcl-2* cell lines. Upregulation of A1 expression in these cells was confirmed by Western blotting (Fig. 7c). Cell competition assays in vitro confirmed that A1-activated cells possessed a striking survival advantage over parental *E*μ-*Myc*/*dCas9a-SAM*^KI/+^/*sgBcl-2* cells in the presence of venetoclax (Fig. 7d). This confirms that upregulation of A1 confers resistance to venetoclax. Similar results were obtained when cells that naturally express MCL-1 were engineered to further upregulate MCL-1 expression levels rather than A1 (Supplementary Fig. 11d, e). In addition to the genes encoding pro-survival BCL-2 proteins, we also identified a number of other genes targeted by the enriched sgRNAs that may be interesting to investigate in the context of venetoclax resistance (Supplementary Fig. 11 and Supplementary Data 2 and 3).

## Discussion

Traditional CRISPR loss-of-function methodologies have become routine in medical research, permitting the identification and validation of tumour suppressor genes and potential therapeutic targets which impact tumourigenesis or sensitivity to anti-cancer therapies[40]. However, identification of oncogenic drivers relies on the enhanced expression of specific gene products and therefore requires different methodologies. While cDNA expression strategies have proven successful for some models, the expression levels achieved with this approach are often variable, leading to non-physiological gene functions, and rarely are multiple transcript isoforms expressed as part of these experiments[9]. In contrast, CRISPRa systems can induce gene expression from the endogenous genomic locus at physiologically or pathologically relevant levels. Of particular interest is the application of CRISPRa to generate faithful models of gene over-expression driven diseases which were previously unattainable due to these limitations. To enable this, we generated a CRISPRa mouse which can be used to activate gene expression in vivo and in vitro. These mice express a dCas9a-

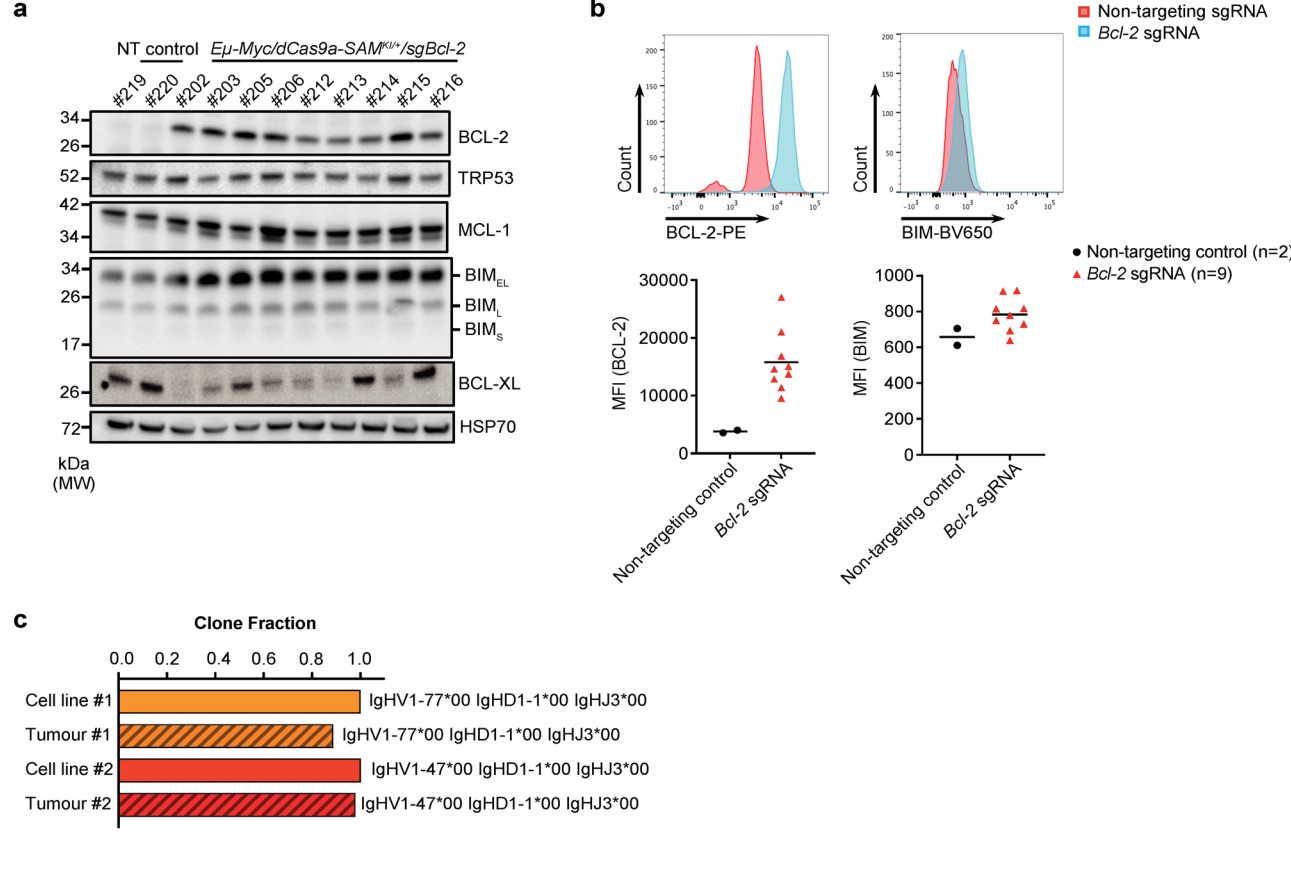

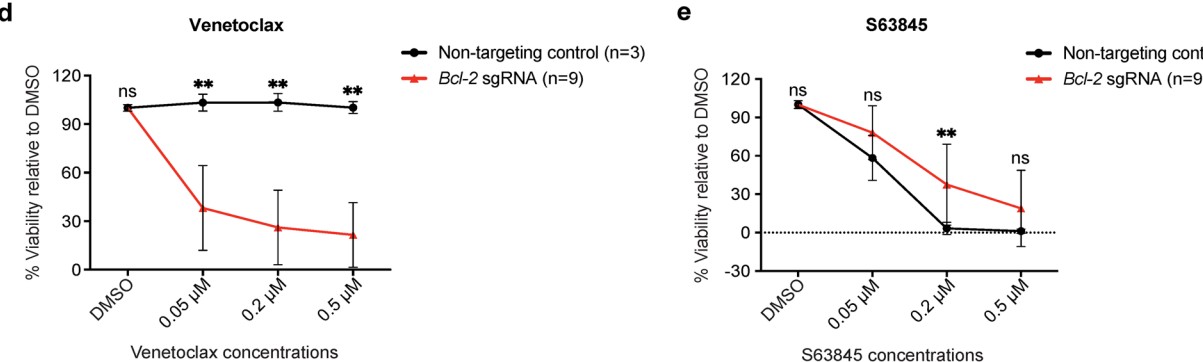

**Fig. 5 | Characterisation of *Eμ-Myc/dCas9a-SAM*^KI/+*/sgBcl-2* lymphoma-derived cell lines. a** The expression of the indicated proteins in control lymphoma-derived cell lines or *Eμ-Myc/dCas9a-SAM*^KI/+*/sgBcl-2* lymphoma-derived cell lines is shown by Western blotting. The indicated numbers of the cell lines represent the animal numbers of the mice that developed lymphoma from which the cell lines had been derived. Probing for HSP70 served as a protein loading control. 2 independent experiments were performed showing similar results. **b** The expression of BCL-2 and BIM in cell lines was determined by intracellular staining and flow cytometry. 2 control cell lines and 9 *Eμ-Myc/dCas9a-SAM*^KI/+*/sgBcl-2* lymphoma cell lines were used for the analysis. Data are represented as the mean MFI of BCL-2 expression (*sgBcl-2*/sgNT) = 15,775/3,796; the mean MFI of BIM expression (*sgBcl-2*/sgNT) =

784/658. **c** Clonotyping of two primary *Eμ-Myc/dCas9a-SAM*^KI/+*/sgBcl-2* tumours and tumour derived cell lines showing monoclonal origin by immunoglobulin heavy chain sequencing using MiXCR analysis of RNA-seq data. **d, e** Cell viability upon treatment with the BCL-2 inhibitor venetoclax or the MCL-1 inhibitor S63845. Control lymphoma cell lines (*n* = 3) and *Eμ-Myc/dCas9a-SAM*^KI/+*/sgBcl-2* lymphoma cell lines (*n* = 9) were treated for 24 h with the indicated concentrations of venetoclax or S63845. Cell viability was determined by propidium iodide (PI) staining followed by flow cytometric analysis. Data are presented as mean ± SD and the statistical significance was evaluated by multiple unpaired *t*-test with two-stage step-up correction. Exact *P* values are provided in the Source Data file. ns = no significant difference, **P* < 0.01. Source data are provided as a Source Data file.

SAM cassette from the *Rosa26* locus for robust expression across different cell types. We selected the SAM system because recent reports suggest that this system is capable of strong transcriptional activity at multiple gene loci[41]. Interestingly, a recent SAM-based *Rosa26* knockin mouse model has been described[42], but our model contrasts in that we employed the CAG promoter because it has been shown to be amongst the most active polymerase II promoters across multiple tissues and cell lines[43]. Using primary cells isolated

from our CRISPRa mouse, we demonstrated induction of expression of B cell specific surface markers in T cells and vice versa. Interestingly, it appears that the CRISPRa mediated induction of already active genes is higher in B cells and genes that are usually silenced can be induced more efficiently in T cells (Fig. 3). This observation could be due to the activities of the sgRNAs or a biological feature of unique cell contexts, which is currently not fully understood. Many more sgRNAs targeting diverse genetic loci need to be tested before

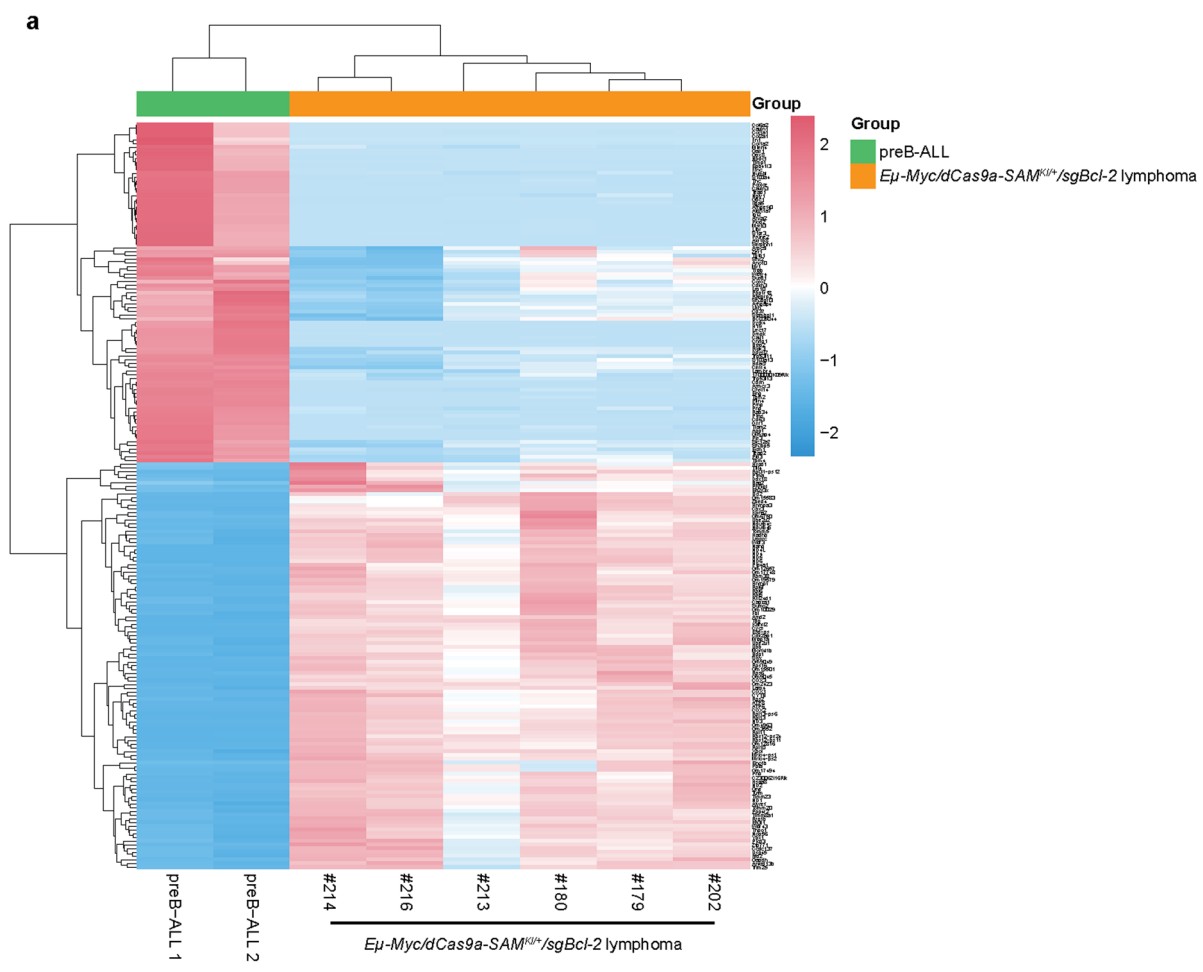

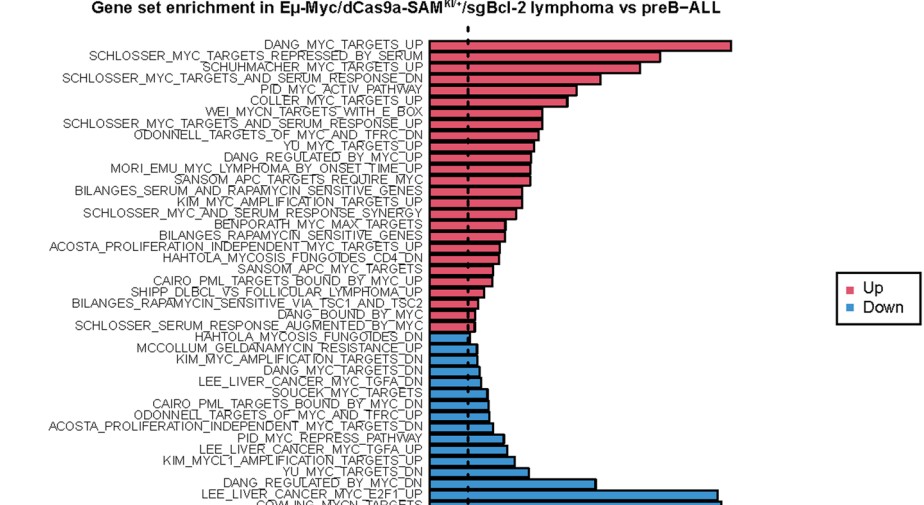

**Fig. 6 | The RNA-seq analysis comparing the genetic patterns of *Eμ-Myc/dCas9a-SAM^{KI/+}/sgBcl-2* lymphomas vs preB-ALL. a** Heatmap of the top 200 most variable genes comparing two preB acute lymphoblastic leukaemia samples (preB-ALL) and six *Eμ-Myc/dCas9a-SAM^{KI/+}/sgBcl-2* tumours demonstrating distinct transcriptional signatures. **b** Gene set enrichment (FDR < 0.05 by *Camera* gene set enrichment) using the Broad Institute MSigDB c2 curated gene sets showing significant enrichment of MYC pathways in *Eμ-Myc/dCas9a-SAM^{KI/+}/sgBcl-2* tumours. Source data are provided as a Source Data file.

**a**

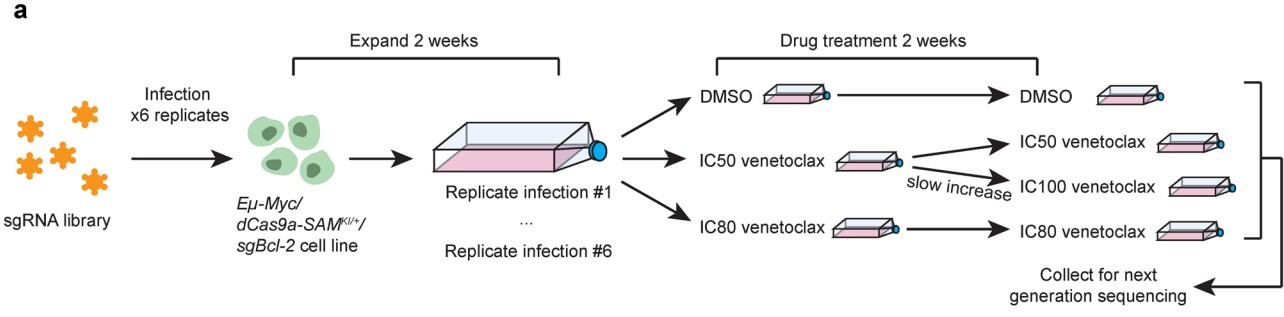

**b**                                                                **c**

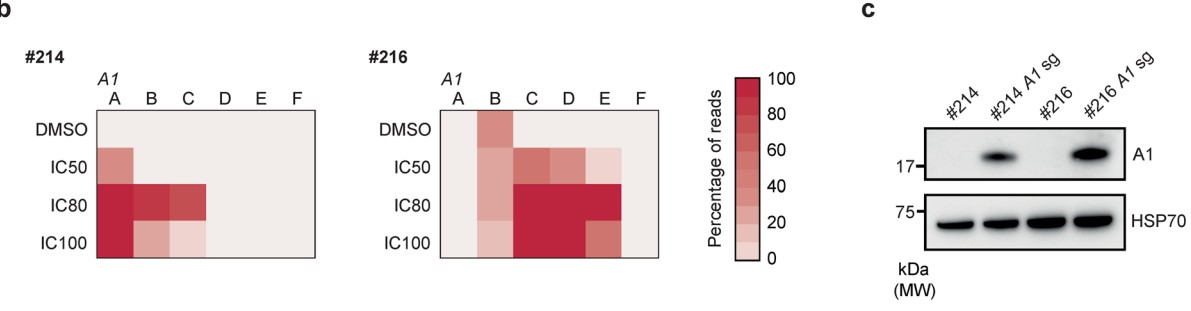

**d**

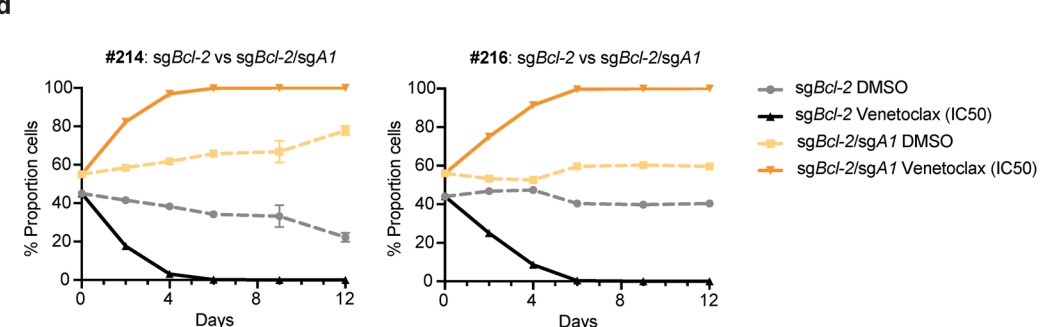

**Fig. 7 | Whole genome CRISPR activation screens in high MYC/BCL-2 expressing lymphoma-derived cell lines reveal upregulation of A1 as a dominant resistance factor for venetoclax treatment. a** Schematic of whole genome CRISPR activation screens performed in two independently derived murine lymphoma cell lines (#214, #216). Six replicates of sgRNA library-infected cells were cultured for 2 weeks in the presence of vehicle (DMSO; control) or the indicated doses of the BCL-2 inhibitor venetoclax before being analysed by next generation sequencing to identify enriched sgRNAs. **b** Heatmaps showing the proportions of sequencing read results which mapped to sgRNAs targeting the promoter regions of the pro-survival gene *Bcl2a1 (A1)*. The six replicate samples for each treatment condition are shown in columns A-F. **c** Western blots confirming upregulation of A1 protein in DHL-like

cell lines transduced with individual sgRNAs targeting the *A1* promoter (sg*A1*). Probing for HSP70 was used as a protein loading control. 2 independent experiments were performed showing similar results. **d** Cell competition assays of #214 or #216 DHL-like cells vs the same cell lines carrying an additional sgRNA targeting *A1* (sg*A1*) as shown in c. Cells were mixed ~1:1 and treated with DMSO (negative control) or IC50 doses of venetoclax. Parental cells were tagged with GFP and the contributions of cells of each genotype to the population were monitored over time by flow cytometry. Data are presented as mean ± SD with representative graph showing one of three experiments performed in duplicate of similar results. Source data are provided as a Source Data file.

further conclusions can be drawn. Regardless, our experiments thus far demonstrate that even silenced genes can be induced with our gene activation mouse model. There are now a number of CRISPR activation models available for the research community[44–46], but since we are unable to compare all of them side-by-side, it is impossible to draw conclusions about effectivity and usefulness of each. We tested expression of our dCas9a-SAM system by Western blotting for dCas9-VP64, which demonstrated wide-spread expression in both haematopoietic and non-haematopoietic tissues. In addition, we showed CRISPR activation of BCL-2 in *dCas9a-SAM^{KI/+}* MEFs. These results strongly suggest activity of the CRISPRa system in diverse cell types. Furthermore, we show that only one copy of our CRISPRa system is able to activate even normally transcriptionally silenced genes, meaning that our *dCas9a-SAM* mice can be crossed

with other mice to enable CRISPR activation in diverse disease models and to this end we will make our mice available to the scientific community for future studies.

We have exploited our robust CRISPRa mouse model to establish a sought after pre-clinical model of aggressive lymphoma highly reminiscent of DHL. This disease is an aggressive form of DLBCL with currently limited treatment options[47]. Thus far, there are no mouse models, besides xenograft models using human DHL cell lines and primary patient samples[16], that recapitulate the key characteristics of this disease. Although previous attempts were made to generate a model of DHL by intercrossing *Eμ-Myc* with *Eμ-Bcl-2* transgenic mice[19], these tumours were identified as immature haematopoietic progenitor tumours with no *Ig* or *Tcr* gene rearrangements and did not represent classical B cell lymphomas. Additionally, these lymphomas could not

be grown in culture, which made it almost impossible to further characterise them or use them to identify drug targets. The BCL-2 over-expressing lymphomas from our mice are committed to the B cell lineage (Fig. 4d), and can readily be grown as cell lines. While it is difficult to compare BCL-2 levels between our lymphomas and the lymphomas from the *Eμ-Myc/Eμ-Bcl-2* double transgenic mice (note human BCL-2 was used in this model), we found similar expression levels by Western blotting. However, BCL-2 was downregulated in the established cell lines in our DHL-like model (Supplementary Fig. 8). Whether this modulation of BCL-2 protein levels contributes to the capability to generate cell lines from these lymphomas or whether it is the cell of origin in which the transformation happens (immature B cell in our DHL-like model (Supplementary Fig. 9), compared to haematopoietic progenitors in the *Eμ-Myc/Eμ-Bcl-2* model[19]) is not clear yet.

Our DHL-like model shows markers of an immature B cell phenotype, i.e. CD43 positive and no increase in germinal centre (GC) B cells (Supplementary Fig. 7, 9), suggesting that the cell of origin is not a GC B cell, which is thought to be the case in human DHL. However, comparing gene expression of our DHL-like lymphomas to immature preB cell ALL murine tumours demonstrated enrichment of pathways associated with human DLBCL in the murine DHL model, in particular, MYC driven pathways including genes associated with DLBCL/DHL subsets (Fig. 6, e.g. SHIPP_DLBCL_VS_FOLLICULAR_ LYMPHOMA)[48]. The results from this pathway analysis along with other features, such as high levels of both MYC and BCL-2 expression, accelerated aggressive tumour onset, positive CD19 expression and monoclonal tumour types, reveals that our murine lymphoma model fulfils many criteria of a classical DHL despite not being of GC origin. We may postulate that the translocations that lead to high BCL-2 and high MYC expression in DHL may arise at the GC B cell stage but that, as evidenced by the RNA-seq analysis, the high expression of MYC/BCL-2 is what drives the tumour signature. Therefore, we believe the DHL-like lymphoma model we have generated will be a very useful model to test therapeutic approaches in vitro, and more importantly in vivo, that are of relevance for agressive human DHL. In this regard, human DLBCL/ DHL xenograft models are useful for investigating drug combinations, but they can be challenging to engraft, even in immune deficient mice. An advantage of our murine DHL-like model is that the tumour cells can be readily transplanted into fully immune competent mice, allowing testing of different therapy approaches in the natural environment.

Most cell lines derived from our murine BCL-2 expressing lymphoma model showed a strong dependency on BCL-2 for their survival[49], which is in striking contrast to conventional *Eμ-Myc* lymphoma cells which are almost exclusively dependent on MCL-1 for their survival[28,29]. Interestingly though, our developed lymphoma cell lines also displayed sensitivity to MCL-1 inhibition. Of note, we identified one BCL-2 expressing lymphoma cell line which is fully resistant to MCL-1 inhibitor treatment, but sensitive to venetoclax. Interestingly, human DHLs were expected to be highly venetoclax sensitive due to their high expression of BCL-2, however, venetoclax monotherapy has only been tested in patients with relapsed DLBCL, including a limited number of DHL patients, which produced low overall response rates[50]. However, these patients were heavily pre-treated and sample sizes were small, highlighting a need for more pre-clinical evidence that venetoclax therapy is worth pursuing for this disease. In this regard, it will be interesting to further understand how the change in pro-survival BCL-2 family member dependency in some of our murine lymphomas developed. As most of the lymphoma cells showed sensitivity to either, or in many cases both, BCL-2 and MCL-1 inhibitors, this raises the possibility that combinations of BH3-mimetics targeting MCL-1 and BCL-2, perhaps with addition of further anti-cancer agents, could have a place in the treatment of patients with DHL. A similar regimen has recently been shown to be effective for the treatment of AML in pre-clinical models[51].

Since recent reports indicated emergence of resistant disease in patients with CLL and AML treated with venetoclax[52,53], we sought to exploit our venetoclax-sensitive murine lymphoma cell lines to identify resistance factors. We took advantage of the CRISPRa machinery already expressed in these cells to perform genome-wide activating library screens. Interestingly, we identified the pro-survival BCL-2 family member A1 (called BFL-1 in humans) as the dominant driver of venetoclax resistance in these cells. Our results expand upon the observation that BFL-1 expression correlates with upfront insensitivity to BH3 mimetics[54] by showing in non-biased CRISPR activation screens that acquired resistance to venetoclax can occur through A1 upregulation. A1 represents an exciting drug target, as deletion of A1 in murine models caused no overt abnormalities[55,56]. Hence, a BH3-mimetic drug targeting A1/BFL-1 is expected to cause no on-target toxicity to healthy cells and could likely be used in combination with venetoclax or MCL-1 inhibitors. Thus far, no direct inhibitors of A1 exist, though CDK9 inhibitors have been reported to reduce A1 expression[54]. However, CDK9 inhibitors also diminish the expression of MCL-1 and many other short-lived proteins; therefore on-target side effects of CDK9 inhibitors are expected to be substantial. Accordingly, clinical trials of BH3-mimetics targeting MCL-1 have recently led to some concerning side-effects[38]. Our results suggest that A1/BFL-1 inhibition may represent an alternative therapeutic approach to overcoming venetoclax resistance in the clinic.

In summary, we have developed a highly efficient CRISPR activation mouse model that can be readily adapted to address both basic biological and translationally-focused research questions across many disciplines.

## Methods

### Animal strains and husbandry
The care and use of experimental animals were performed in accordance with the rules and guidelines set out by the WEHI Animal Ethics Committee. The laboratory mice were kept under the SPF condition and were maintained on a 12-h light/dark cycle at the controlled room temperature of 20–22 °C with humidity of 40–50% during experiments. *dCas9a-SAM[KI]* mice were generated on a C57BL/6 genetic background using CRISPR techniques as previously described[57]. *Eμ-Myc* transgenic mice have been described previously[26]. C57BL/6-Ly5.1 mice were obtained from stocks held at WEHI.

### Construct cloning
All single guide RNA (sgRNA) sequences used for the CRISPR activation system were designed using the sgRNA design tool developed by the Zhang Lab (https://portals.broadinstitute.org/gppx/crispick/public). The sgRNA vectors (LV06, CRISPRa SAM U6-gRNA: EF1a-Puro) targeting the promoter region of mouse *Bcl-2* or *Mdm2* genes and non-targeting control sgRNA vectors were purchased from Sigma–Aldrich. The puromycin resistance gene within the SAM sgRNA vector was replaced by blue fluorescent protein (BFP) coding sequences to generate a SAM sgRNA vector that expresses BFP (CRISPRa SAM U6-gRNA: EF1a-BFP). The sgRNAs targeting the promoters of the mouse *Cd4, Cd19* and *Irf4* sgRNAs gene promoters were cloned into this vector. The sgRNA sequences are shown in Supplementary Table 3.

### Cell culture
All cell lines were cultured at 37 °C with 10% $CO_2$ in a humid atmosphere. The human embryonic kidney cell line expressing the SV40 large T antigen (293 T, American Type Culture Collection, #CRL-3216) was maintained in Dulbecco's modified Eagle's medium (DMEM, Gibco, #11995065) containing 10% (v/v) heat-inactivated Foetal Bovine Serum (HI-FBS; Sigma–Aldrich, #12007 C) and 100 U Penicillin-Streptomycin solution (Pen-Strep, Sigma–Aldrich, #P0781).

The *Eμ-Myc* lymphoma-derived cell lines EMRK-1184 and MRE-721 were generated in our laboratory. The *Eμ-Myc/dCas9a-SAM[KI/+]/sgBcl-2*

(or *sgMdm2*, or sgNT) lymphoma cell lines were derived from tumour tissues of mice that had been lethally irradiated and then transplanted with *Eμ-Myc/dCas9a-SAM$^{KI/+}$* HSPCs that had been transduced with *Bcl-2* or *Mdm2* targeting, or non-targeting control sgRNAs. All *Eμ-Myc* lymphoma cell lines were cultured in DMEM, 10% (v/v) HI-FBS, 23.8 mM sodium bicarbonate (Sigma–Aldrich, #S8761), 1 mM HEPES (Gibco, #15630080), 13.5 μM folic acid (Sigma–Aldrich, #F8758), 0.24 mM L-asparagine monohydrate (Sigma–Aldrich, #A7094), 0.55 mM L-arginine mono-hydrochloride (Sigma–Aldrich, #A6969), 22.2 mM D-glucose (Ajax Finechem, #713), 100 U-μg/mL Pen-Strep and 50 μM 2-mercaptoethanol (2-ME, Sigma-Aldrich, #M3148). This medium is referred to as FMA.

### Lentiviral production and target cell infection
The lentiviral packaging system containing 5 μg p-MDL, 2.5 μg p-RSV-REV, 3 μg p-VSVG and 10 μg target DNA vector was transduced into 293 T cells using the calcium phosphate transfection method[58]. Viral supernatants along with 8 μg/mL polybrene were added to $1 \times 10^5$ *Eμ-Myc* lymphoma cells and centrifuged at 1100 x *g* for 2 h at 32 °C. After aspiration of viral supernatants, target cells were resuspended in FMA medium and seeded onto 12-well plates.

For infection of HSPCs, the lentiviral packaging system containing 5 μg p-MDL, 2.5 μg p-RSV-REV, 5 μg p-ECO envelope (ENV) and 10 μg target DNA vector was transduced into 293 T cells using the calcium phosphate transfection method[58]. Viruses were used to infect HSPCs derived from E14.5 foetal livers or activated B cell blasts or T cell blasts derived from splenocytes.

### Cell viability assay
Cell viability assays were performed on *Eμ-Myc* lymphoma cell lines to assess sensitivity to BH3 mimetic drug treatment. Cells were plated into 96-well flat-bottom plates at a density of $3 \times 10^4$ cells per well and treated for 24 h with the MCL-1 inhibitor S63845 (Active Biochem, #A-6044) or the BCL-2 inhibitor venetoclax (Active Biochem, #A-1231) at the indicated concentrations. Cell viability was determined by staining with 1 μg/mL propidium iodide (PI, Sigma–Aldrich, #P4864) followed by flow cytometric analysis using the LSR-II Analyzer (BD Biosciences). Flow cytometry data were analysed using FlowJo v10 software (BD Biosciences).

### Long range PCR and Southern blotting
The genomic DNA of WT or *dCas9a-SAM$^{KI/+}$* or *dCas9a-SAM$^{KI/KI}$* mice was extracted from mouse liver using a DNeasy Blood & Tissue Kit (QIA-GEN, #69504). Long range PCR (LR-PCR) and Southern blot were performed to verify the insertion of the dCas9a-SAM cassette into the mouse *Rosa26* locus. Each LR-PCR reaction was performed with 100 ng of DNA using the PrimeSTAR GXL polymerase (TaKaRa Bio, #R050B). The conditions used for LR-PCR were: 95 °C for 2 min, (95 °C for 30 sec, 68 °C for 5 min) x 30 cycles, 68 °C for 5 min. The amplified products were analysed by agarose gel electrophoresis. Primers used for LR-PCR were FWD: 5′-GCCGACGCTAATCTGGACAAAGTGCTG-3′ and REV: 5′-ACATTACTGTCACTGACCATCATGCCTCTG-3′.

The genomic DNA used for Southern blotting was extracted from livers of WT or *dCas9a-SAM$^{KI/+}$* or *dCas9a-SAM$^{KI/KI}$* mice using proteinase K lysis buffer followed by phenol/chloroform purification. 40 μg of each DNA sample was digested with EcoRV and RNase overnight. DNA fragments were electrophoresed and transferred onto a nylon membrane. The combined filter was placed in a Stratalinker and UV crosslinked with 1200 kJoules (x100; standard autocrosslink setting). P32-labelled 3′ and 5′ *Rosa* probes were prepared using the DECAprime II DNA labelling kit (Invitrogen, #AM1456). The filter was pre-hybridised for 30 min and hybridised with probes overnight at 42 °C. The filter was then washed and exposed on a phosphorimager overnight.

### Haematopoietic cell analysis and flow cytometry
Thymus, bone marrow, spleen and lymph node tissues were harvested from WT or *dCas9a-SAM$^{KI/KI}$* mice. Detailed information on specific mice is displayed in Supplementary Table 4. Tissues were homogenised using a syringe plunger and passed through a 70 μM cell strainer to generate single cell suspensions. Single cells were then washed and resuspended in PBS (Gibco, #14190144) containing 5 μM EDTA (Sigma–Aldrich, #E8008) and 5% (v/v) HI-FBS. Cell counts were determined using the Moxi Z Mini Automated Cell Counter (ORFLO Technologies). Intracellular staining was performed using the Intracellular Fixation & Permeabilization Buffer Set (eBioscience, #88-8824-00) according to the manufacturer's instructions. All fluorochrome conjugated antibodies used for extracellular or intracellular staining are listed in Supplementary Table 5. Staining with PI (1 μg/mL) was used to exclude dead cells. Fluorescence was quantified using the LSR Fortessa X-20 Cell Analyzer (BD Biosciences) via BD FACSDiva software v8.0 and data were analysed using FlowJo v10 software. All flow cytometry analytical gatings are shown in Supplementary Fig. 12.

### Haematopoietic cell reconstitution experiments
Female *dCas9a-SAM$^{KI/KI}$* mice were crossed with *Eμ-Myc* male mice to produce *Eμ-Myc/dCas9A$^{KI/+}$* embryos. *Eμ-Myc/dCas9a-SAM$^{KI/+}$* HSPCs were isolated from foetal livers of E14.5 embryos and maintained in foetal liver medium containing α-MEM GlutaMAX (Gibco, #32561037), 10% (v/v) HI-FBS, 100 U-μg/mL Pen-Strep, 10 mM HEPES, 1 mM L-glutamine (Gibco, #25030081), 1 mM sodium pyruvate (Gibco, #11360070), 50 μM 2-ME, and supplemented with recombinant mouse cytokines made in house. The cytokines used for HSPC culture were 100 ng/mL mouse stem cell factor, 10 ng/mL mouse interleukin-2, 50 ng/mL mouse thrombopoietin and 10 ng/mL mouse Fms-like tyrosine kinase 3. All cytokines were produced and kindly provided by Dr. Jian-Guo Zhang (WEHI).

Supernatants containing ecotropically packaged lentiviruses expressing sgRNA were added onto retronectin-coated 12-well plates along with 8 μg/mL polybrene and centrifuged at 2700 × *g* for 90 min at 32 °C. After aspiration of viral supernatants, cultured HSPCs were seeded onto each well supplemented with 0.5 mL fresh medium containing cytokines and incubated at 37 °C overnight. The virus-infected HSPCs were then injected intravenously (i.v.) into lethally irradiated (2 × 5.5 Gy, 4 h apart) C57BL/6-Ly5.1 recipient mice (female, 6-week old). Reconstituted mice were monitored for lymphoma development and overall appearance. Survival time was defined as the time from HSPC transplantation until reconstituted mice had to be sacrificed due to reaching the predefined ethical endpoint determined by experienced animal technicians at WEHI Bioservices. Signs of lymphoma include increased respiration, palpable enlarged lymphatic organs, hunched posture and general loss of condition. Peripheral blood and tumour tissues collected from mice at the ethical endpoint were used for Western blot analysis, generation of cell lines and flow cytometric analysis.

### RNA preparation and RNA-seq analysis
Mouse Embryonic Fibroblast (MEF) cell lines were generated from E14 embryos of WT or *dCas9a-SAM$^{KI/+}$* mice and cultured in DMEM containing 10% (v/v) HI-FBS, 100 U-μg/mL Pen-Strep and 1 mM L-glutamine. For RNA-seq experiments, *dCas9a-SAM$^{KI/+}$* MEFs were infected with non-targeting control sgRNAs.

Cells were lysed in TRIzol Reagent (Thermo Fisher Scientific, #15596026) and RNA was extracted using the Direct-zol RNA Microprep kit (Zymo Research, #R2061) including DNase digestion step according to the manufacturer's instructions. RNA samples were prepared for sequencing using the TruSeq RNA Library Prep Kit v2 (Illumina, #RS-122-2001) according to the manufacturer's

instructions. The prepared library was sequenced on a NextSeq 1000 (Illumina).

## Culture and infection of primary immune cells

Naïve B cells were purified with Streptavidin Negative Selection Beads (Invitrogen, #MSNB-6002) by staining with CD4, CD8, TER119 and CD43 cell surface markers. B cells were cultured in RPMI 1640 medium supplemented with 10% (v/v) HI-FBS, 100 U·µg/mL Pen-Strep, 1% L-glutamine, 1% non-essential amino acids (Gibco, #11140050), 1% HEPES, 1% sodium pyruvate and 50 µM 2-ME. T cells were cultured in FMA medium supplemented with 10 ng/mL mouse interleukin-2. Naïve B cells and T cells were stimulated with 10 µg/mL LPS (B cells) or 2 µg/mL concanavalin A plus 10 ng/mL mouse interleukin-2 (T cells), respectively.

For lentiviral infection, $5 \times 10^5$ activated B cells per well were added onto retronectin-coated 12-well plates. Lentiviruses expressing sgRNA were added onto cells along with 8 µg/mL polybrene and centrifuged at $300 \times g$ for 90 min at 28 °C. After aspiration of viral supernatants, fresh medium was added into each well and cells were incubated under normal conditions. Lentiviral supernatants along with 8 µg/mL polybrene were added to $1 \times 10^6$ activated T cells and centrifuged at $1100 \times g$ for 2 h at 32 °C. After aspiration of viral supernatants, T cells were resuspended in fresh medium and seeded onto 12-well plates.

## Western blotting

Liver, kidney and heart were harvested from WT or *dCas9a-SAM*[KI/KI] mice. 0.03 g of each tissue was dissected and added into 100 µL ice-cold RIPA lysis buffer (50 mM Tris-HCl (pH 8.0), 150 mM NaCl, 1% NP-40, 0.5% Sodium Deoxycholate, 0.1% SDS) containing EDTA-free protease inhibitors (Roche, #04693116001). Tissues were homogenised using a metal probe with an additional 300–600 µL of ice-cold lysis buffer added during the homogenization. Lysates were rotated for 2 h at 4 °C and then centrifuged at $12,000 \times g$ for 20 min at 4 °C to collect supernatants containing protein. For cell samples, cell pellets were collected and resuspended in RIPA lysis buffer containing protease inhibitors and incubated on ice for 15 min. Centrifugation was performed at $12,000 \times g$ for 10 min at 4 °C to collect supernatants containing protein.

Protein concentrations were measured using the BCA Protein Assay Kit (Thermo Fisher Scientific, #23225) following the manufacturer's protocols. For Western blotting, a total of 15 µg protein was loaded onto NuPAGE 4–12% Bis-Tris 1.5 mm gels (Life Technologies, #NP0335BOX) and proteins were size fractionated by gel electrophoresis. The iBlot 2 Dry Blotting System (Life Technologies, #IB23001) was used to transfer proteins onto 0.2 µm nitrocellulose membranes. Membranes were blocked in 5% (m/v) skim milk and incubated with primary antibodies overnight at 4 °C. Primary antibodies used for Western blotting are listed in Supplementary Table 6. Membranes were washed and incubated with appropriate horseradish peroxidase (HRP)-conjugated secondary antibodies to detect mouse, rat or rabbit IgG (Southern Biotech, #1010-05, #3010-05, #4010-05). Antibody bound proteins on membranes were visualised by adding Luminata Forte Western HRP substrate (Millipore, #WBLUF0100) and imaging was performed using the ChemiDoc XRS + machine (Bio-Rad).

## CRISPR activation screening

Whole genome CRISPR/Cas9 activating screens were performed in *Eµ-Myc/dCas9a-SAM*[KI/+]*/sgBcl-2* DHL-like derived cell lines. To prepare virus containing the sgRNA library, 10 µg Caprano CRISPRa sgRNA library (A and B combined)[39] was transfected into 293 T cells along with 5 µg p-VSVG and 10 µg psPAX2 as described above. *Eµ-Myc/dCas9a-SAM*[KI/+]*/sgBcl-2* lymphoma-derived cell lines (#214, #216), which are highly sensitive to venetoclax, were split into 6 replicates of $3 \times 10^5$ cells each and infected as described above for the *Eµ-Myc* lymphoma

cells. Infected cells were expanded in culture for 2 weeks. For each replicate infection, $7 \times 10^6$ cells were transferred into three T75 flasks and treated with either DMSO (negative control), or IC50 doses (214: 10 nM, 216: 5 nM) or IC80 doses (214: 30 nM, 216: 15 nM) of venetoclax. Lymphoma cells were passaged every 3–5 days as they became confluent and fresh drug was added. Cells receiving the IC50 dose of venetoclax were split into two other T75 flasks after 4 days of treatment. One culture was maintained at an IC50 dose while the dose for the other was ramped up slowly over time until an IC100 dose (214: 100 nM, 216: 50 nM) was reached. After a total of 14 days of drug treatment, pellets of $2 \times 10^6$ cells were collected for all treatments. DNA was extracted using a DNeasy Blood & Tissue Kit. sgRNAs were amplified from 100 ng of DNA using GoTaq Green Master Mix (Promega, #M7123) according to the manufacturer's protocol. The following primers were used, which had been modified with unique overhangs to create a set of indexing primers for Illumina sequencing: FWD: 3′-TGCTTACCGTAACTTGAAAGTA-5′ and REV: 5′-AATACGAG-CAGACCCTGATG-3′. PCRs were performed in triplicate for each sample. Products were pooled, cleaned up using Ampure XP beads (Beckman Coulter, #A63881) and sequenced on an Illumina NextSeq 550. For each sample, the number of reads mapping to each sgRNA in the library (as a proportion of the total number of reads for that sample) was calculated. For bar graphs, sgRNAs which made up at least 10% of the total sgRNAs detected for each sample were plotted.

## Validation of hits from the CRISPR screens

To validate hits identified from CRISPR screens, select sgRNA sequences from the Caprano library were individually cloned into pXPR_502. The sgRNA sequences are shown in Supplementary Table 3. Virus was prepared and used to transduce $1 \times 10^5$ *Eµ-Myc/dCas9a-SAM*[KI/+]*/sgBcl-2* lymphoma-derived cells as described above. Infected cells were cultured for 2–3 weeks to allow time for gene induction. Meanwhile, parental *Eµ-Myc/dCas9a-SAM*[KI/+]*/sgBcl-2* lymphoma-derived cell lines were transduced with a retroviral vector expressing high eGFP (pMIG) as described above, using 10 µg pMIG, 5 µg GAG and 5 µg ENV. eGFP-high cells were sorted on a FACSAria III flow cytometer (BD Biosciences). eGFP-high parental cells were mixed 1:1 with cells carrying the sgRNAs for validation. Mixed cultures were treated with DMSO (negative control) or an IC50 dose of venetoclax for up to 2 weeks. Cultured cells were passaged every 2–3 days, fresh drug added, and the proportions of each cell population monitored by flow cytometry.

## Statistical analysis

All data were analysed by Graphpad Prism software (Version 8.2.0). The comparison between two groups was determined by student's *t*-test. For more than two groups, multiple *t*-tests or two-way ANOVA were performed to evaluate differences between the groups. Kaplan-Meier survival curves were plotted to represent the tumour latency of HSPC reconstituted mice. Data are represented as means ± the standard deviation (SD) and significance between groups are measured by *P* value that is statistically significant as *$P < 0.05$, **$P < 0.01$ and ns = no significant difference.

RNA-Seq analysis of wildtype (WT) and *dCas9a-SAM*[KI/+] MEF and B-ALL samples were analysed using R version 4.1.3 software packages edgeR v3.34.1[59,60] and limma v3.48.3[61]. First, samples were aligned by Rsubread v2.6.4[62] and annotated to Mus musculus genome assembly GRCm38 (mm10) from Genome Reference Consortium. The counts were filtered by using the method described[63] and then transformed to $\log_2$ (counts per million) with associated precision weights, using the voom method[64]. Differential enrichment between treatments, *Eµ-Myc/dCas9a-SAM*[KI/+]*/sgBcl-2* lymphomas vs preB-ALL and WT vs *dCas9a-SAM*[KI/+]*/sgNT* MEFs, were all assessed using empirical Bayes moderated *t*-statistics[65]. The *P* values were adjusted to control the false discovery rate (FDR) using the method of Benjamini and Hochberg[66]. The pathway analyses are conducted using Molecular Signatures Database

v7.5.1 c2 curated dataset[67] using camera[68] gene set test. For visualising gene expressions with heatmaps, we used pheatmap v1.0.12 R package and for visualising gene set enrichments we used barcode plots from limma package. For clonotype analysis, fastq files from cell lines and primary tumours were analysed using the MiXCR software package (Version 3.0.6)[69].

For CRISPR screens, differential enrichment analyses were conducted using edgeR software package edgeR v3.30.3[59,60]. sgRNAs were filtered by using the method described and library sizes were normalised using upperquartile method[70]. Counts were then transformed to $\log_2$ (counts per million) with associated precision weights, using the voom method. Differential enrichment between treatments was assessed using empirical Bayes moderated $t$-statistics. The $P$ values were adjusted to control the FDR using the method of Benjamini and Hochberg.

### Reporting summary

Further information on research design is available in the Nature Research Reporting Summary linked to this article.

## Data availability

RNA-seq data generated in this study have been deposited in the NCBI Gene Expression Omnibus (GEO) database under accession code GSE205509. The Molecular Signatures Database used in this study is available online [https://www.gsea-msigdb.org/gsea/msigdb]. The remaining data are available within the Article, Supplementary Information or Source Data file. Source data are provided with this paper.

## Code availability

All original code used to analyze data reported in the paper are provided at the GitHub repository [https://github.com/goknurginer/crispra-screen-analysis][71].

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

## Acknowledgements

We thank C Gatt, C Epifanio and G Siciliano for expert animal care; B Helbert and K Mackwell for genotyping; S Monard and his team for help with flow cytometry; MA Anderson and D Hilton for insightful discussions and all the members of the Herold and Kelly laboratories. This work was supported by grants and fellowships from the Australian National Health and Medical Research Council (NHMRC) (Project Grants 1159658, 1186575 and 1145728 to M.J.H., 1143105 to M.J.H. and A.S., 1144905 to S.L.N., Ideas Grants 2002618 and 2001201 to G.L.K., Program Grant 1113133 to A.S. and J.E.V., 1113577 to W.S.A., Fellowships 1058344 to W.S.A., 1102742 to J.E.V., 1020363 to A.S., 1156095 to M.J.H. and Investigator Grant 1173342 to W.S.A.), the Leukemia and Lymphoma Society of America (LLS SCOR 7015-18 to A.S., G.L.K., M.J.H.), the Cancer Council of Victoria (project grant 1147328 and 2021 Grant In Aid to M.J.H., 1052309 to A.S., 1147328 to G.L.K. and Venture Grant to M.J.H. and A.S.), Victorian Cancer Agency (MCRF Fellowship 17028 to G.L.K., Leukaemia Foundation of Australia (grant to A.S. and G.L.K.), Phenomics Australia (to A.J.K. and M.J.H.) the estate of Anthony (Toni) Redstone OAM (A.S. and G.L.K.), the Craig Perkins Cancer Research Foundation (G.L.K.), the Dyson Bequest (G.L.K.), the Harry Secomb Trust (G.L.K.) as well as by operational infrastructure grants through the Australian Government Independent Research Institute Infrastructure Support Scheme (361646 and 9000220) and the Victorian State Government Operational Infrastructure Support Program.

## Author contributions

Y.D., S.T.D. performed and planned all experiments, M.A.P., S.T., A.N., G.H., S.R.K., A.C., K.B., A.J.K., M.P., S.W., L.T., G.G. contributed to some of the experiments or provided experimental data, A.H., W.S.A., J.E.V.,

S.L.N., A.S., B.H., Q.Z. gave critical advice, G.L.K. and M.J.H. designed and supervised all experiments. All authors were involved in writing the manuscript.

## Competing interests

The Walter and Eliza Hall Institute receives milestone and royalty payments related to venetoclax. A.S. is consulting for Genentech and Servier. A.H. and B.H. are employees of Genentech, Inc. and shareholders of Roche. The remaining authors declare no competing interests.
