## [Peer Review File · Nature Communications]

Title: A novel CRISPR activation mouse enables modelling of aggressive lymphoma and interrogation of venetoclax resistance.REVIEWER COMMENTS

Reviewer #1 (Remarks to the Author):

In this manuscript, Deng et al., generated a dCas9a-SAMKI mouse model and demonstrated this model could be used to transactivate gene expression in lymphocytes both in vitro and in vivo. They developed a B-cell lymphoma model by crossing this dCas9a-SAMKI strain with Eu-Myc strain and transducing sgBcl-2. They also performed a genome-wide CRISPR-Cas9 activation screen in the DHL-derived lymphoma cells and identified a dominant role for BCL-2 family protein A1 (BFL-1) in venetoclax resistance. Overall, the authors generated a useful dCas9a-SAMKI mouse model. Combined with the potential establishment of a transgenic DHL model (see below) and the whole genome screen make the manuscript a good candidate for publication in Nature Communications. I have a few reservations, as outlined below:

Major comments

1. The primary concern is that the authors have not clearly demonstrated that they generated DHL. There is no histology, immunophenotyping or gene expression data to confirm that the disease is DLBCL-like. These are absolutely required. It would also be quite helpful to have gene expression data from RNAseq to demonstrate similar expression patterns to DHL rather than Burkitt or B-ALL. Finally, the authors have not clarified whether the lymphomas are monoclonal, oligoclonal or polyclonal. There is no mutation analysis or IGH sequencing. This is true for both the in vivo lymphomas and the cell lines, which could also be oligo/polyclonal.
2. The authors state that they have made the first DHL model. Even if they can prove that their models are DHL, the statement is not true. Both humanized and PDX models of DHL have been published by groups in Boston.
3. The authors state that their unique lymphoma phenotype likely arises from lower expression of BCL2 compared to the Eu-Myc/Eu-Bcl2 models but they don't show any data to support it. The Eu-Myc/Eu-Bcl2 should be included as a comparator for the immunoblots (e.g. in Figures 1b, 3a, 4e, 5a).
4. The replicates of genome-wide CRISPR-Cas9 screen are quite heterogeneous, as shown in Fig 6 and supplementary Fig 5. A statistical analysis should be included. The authors report enrichment of sgRNA upregulating of BCL-XL and MCL-1; however, the reads were only upregulated in very few replicates (supplementary Fig 5b). Further explanation is needed. Also, all of the CRISPR screen primary data, including all genes screened, should be included as a Supplementary Table.
5. The same is true for the MCL1 and BCL2 inhibitor data across cell lines. A supplementary table should be added with IC50 values so the reader can compare across lines.

Minor comments

1. In Fig. 1b, the control in lane 5 should be treated with puromycin instead of no puromycin treatment?

And there is a bubble covering the data in lane 5 for BCL-2. The authors should provide a clear result.

2. Fig. 2b lacks sample information/conditions.

3. The controls in Fig 3b-f are questionable. The authors used Cd4-sgRNA as a negative control for Cd19 sgRNA and vice versa. The authors should use non-targeting sgRNA as a control, as they did in Fig 3a. What's more, the authors need to provide the statistical information and present the data (as in Fig 3c) using a dot column for all of the experiments, including Fig 4b, Fig 4d and Fig 5b.

4. There is no information in the figure legend or the methods on mouse sample collection methods, including mouse ages and other details.

5. The authors should provide both vendors and catalog numbers of the reagents used in the study, for example, in the 'Cell culture' section

Reviewer #2 (Remarks to the Author):

The paper presented by Deng and colleagues describes a novel mouse model which allows gene activation by Cas9 expression from the Rosa26 locus and applies the in vivo system to gain insights in aggressive B cell lymphoma pathogenesis.

Overall, the presented data is convincing and the manuscript is well written. However, the following limitations apply:

Major concerns:

The dCas9a-SAMKI mouse generated by Deng and colleagues shares similarities to the CRISPR-activator mouse model published recently by Hunt et al. (Hunt, C., Hartford, S.A., White, D. et al. Tissue-specific activation of gene expression by the Synergistic Activation Mediator (SAM) CRISPRa system in mice. *Nat Commun* 12, 2770 (2021)). The authors might want to discuss the pros/cons of the different animal strains in their manuscript.

The dCas9a-SAMKI mouse will be of interest for researchers in various disciplines. Thus, the proper targeting of the Rosa26 locus is essential to avoid unwanted side-effects.

From the results in Fig 2B it is not clear whether the correct integration of the targeting cassette exclusively in the Rosa26 locus is validated. Additional experiments (Southern blot?) or the detailed description of the PCR approach are needed.

Several analyses demonstrating the efficacy of the modified CRISPR approach for gene activation in B and T cells are shown (see Fig 3). In the reported experiments cells isolated from homozygous transgenic mice (dCas9a-SAMKI/KI) were used.

In the future, the novel dCas9a mouse might be used in combination with other transgenic mouse

strains, in a plethora of them the Rosa26 locus is modified as well. The analysis of heterozygous transgenic mice (dCas9a-SAMKI/+) would be most interesting -> how much dCas9 is expressed from a single allele? Is the amount of dCas9a still sufficient for gene activation (of active and/or silenced genes)?

The B cell lymphomas arising in Emu-MYC, dCas9a-SAMKI/+, BCL2 mice are poorly characterized. As CD19 and B220 co-expression are not restricted to mature B cells, the origin of the tumors is still obscure. Detailed analyses of the tumor cohorts (morphology/histology; proliferation; flow cytometry of B cell differentiation markers, e.g. IgM, IgD, CD43, AA4.1, CD38, FAS; SHM analysis, etc.) are critical to judge the value of the model.

In addition, human Double-hit-lymphomas (DHL) are an aggressive subgroup of DLBCL. Thus, antigen-activated B cells and the GC reaction are the origin of malignant transformation. Unfortunately, data about this cell population in dCas9a-SAMKI animals and in pre-malignant Emu-MYC, dCas9a-SAMKI/+, BCL2 mice are missing.

In DHL MYC and BCL2 are typically affected by genetic rearrangements which bring the oncogenes under the control of "foreign" promoter/regulator elements. Thus, the authors' selection of studying DHL pathogenesis for validating their dCas9 approach (in which gene expression from the endogenous context is achieved) is unclear.

Minor concerns:

In B cells the dCas9a-based activation of already active genes is more efficient than in T cells (for silenced genes, opposite results are demonstrated). In the discussion (biological?) explanations for these findings can be discussed.

Absolute numbers are missing in Fig S2, although this data is stated in the text (Page 6, line 166).

Reviewer #3 (Remarks to the Author):

Deng and Diepstraten et al describe development of a new CRISPRa mouse which they utilize to create a new mouse cancer model of double hit MYC+/BCL2+ lymphomas and then subsequently to better understand tumor response to a BCL2 inhibitor, venetoclax. The authors show this CRISPRa mouse is useful and should be widely adopted. A big part of the impact of this paper will be through adoption of this mouse model. As such it is critically important for this mouse to be made available through Jackson labs or some other means. The authors would strengthen this manuscript by showing CRISPRa proteins are expressed and active in cell types or tissues beyond the hematopoietic system.

The authors should discuss why this new SAM CRISPRa mouse is different or better than other CRISPRa

mice such as PMID: 29335603, 30545847 or preprint: doi: <https://doi.org/10.1101/2021.03.08.434430>. One way to discuss this could be in the context of: PMID: 27214048.

In the introduction the authors discuss shortcomings of previous double hit lymphoma mouse models. The authors should consider citing efforts to create humanized models of double hit lymphomas which do express B cell markers and have been useful for understanding response to anti-cancer therapies (PMID: 22484426 and 24485462). The authors claim that the Eumyc/sgBCL2 tumors are a resource for understanding double hit lymphoma biology and drug response but they need to justify why this model is better than the human double hit model above. One justification could relate to the fact you can use the Eumyc/sgBCL2 model in a fully immunocompetent syngeneic mouse which is not possible with the humanized model.

The authors state that the SAM mouse does not have developmental issues or overt disease which is great but this analysis could be more nuanced. For example, recently analysis of a dCas9-p300 mouse (<https://doi.org/10.1101/2021.03.08.434430>) has revealed off target activity when looking at RNA-seq or ChIP-seq data comparing saline injected mice to sgRNA expressing mice. Do you see any transcriptional differences +/- sgRNA expression? The comparison of saline to a negative control sgRNA via RNA-seq would illuminate possible off target activity. This is important as this mouse also represents a potential resource for others in the field and so it is important to define upfront potential strengths and problems.

Related to Figure 4,5- do the Eu-myc/sgBCL2 lymphomas develop additional secondary mutations in the TP53 pathways as is frequently the case in the Eumyc model? I ask because the comparison to nontargeting control lymphomas will often be a comparison to a lymphoma that has lost TP53, Arf or overexpressed MDM2 according to the older literature.

The observation that BFL-1 upregulation promotes resistance to venetoclax is really interesting as this gene is understudied. It would greatly strengthen this observation if there is patient data for double hit lymphomas treated with venetoclax to support this observation.

For ease of reading, the comments from the reviewers are presented in black italics; and our responses are presented in plain blue script.

REVIEWER COMMENTS

Reviewer #1 (Remarks to the Author):

In this manuscript, Deng et al., generated a dCas9a-SAMKI mouse model and demonstrated this model could be used to transactivate gene expression in lymphocytes both in vitro and in vivo. They developed a B-cell lymphoma model by crossing this dCas9a-SAMKI strain with Eu-Myc strain and transducing sgBcl-2. They also performed a genome-wide CRISPR-Cas9 activation screen in the DHL-derived lymphoma cells and identified a dominant role for BCL-2 family protein A1 (BFL-1) in venetoclax resistance. Overall, the authors generated a useful dCas9a-SAMKI mouse model. Combined with the potential establishment of a transgenic DHL model (see below) and the whole genome screen make the manuscript a good candidate for publication in Nature Communications. I have a few reservations, as outlined below:

We thank the reviewer for their positive comments about our manuscript and we have addressed their suggestions for changes as detailed below.

Major comments

1. The primary concern is that the authors have not clearly demonstrated that they generated DHL. There is no histology, immunophenotyping or gene expression data to confirm that the disease is DLBCL-like. These are absolutely required. It would also be quite helpful to have gene expression data from RNAseq to demonstrate similar expression patterns to DHL rather than Burkitt or B-ALL. Finally, the authors have not clarified whether the lymphomas or monoclonal, oligoclonal or polyclonal. There is no mutation analysis or IGH sequencing. This is true for both the in vivo lymphomas and the cell lines, which could also be oligo/polyclonal.

We thank the Reviewer for the comment, and we have performed further analysis on pre-leukaemic mice as well as fully malignant lymphoma samples. In the pre-leukaemic analysis, we found overall many more B cells in mice transplanted with Eu-Myc/dCas9a/sgBCL-2 HSPCs as compared to the control mice. These B lymphoid cells were mostly naïve B cells, and we did not see any increase in germinal centre (GC) B cells. Analysis of the established lymphoma cell lines revealed an immature (CD43+) surface Ig positive and negative phenotypes present. These findings led us to the conclusion that the cells of origin (COO) for our lymphomas are B cell committed but are not GC B cells and we would therefore not expect any mutations of the IgV coding regions.

As also suggested by Reviewer #1 we performed RNAseq analysis of our lymphomas and compared them to a B-ALL RNAseq data set available at WEHI. We find clear differences between the Eu-Myc/dCas9a/sgBCL-2 lymphomas and B-ALL lymphomas, including up-regulation of genes and pathways associated with DLBCL, including those perturbed with DHL translocations (Figure 6 and Supplementary Table 2). Analysis of Ig heavy chain type sequence clearly established our lymphomas are monoclonal, like human DHLs (Figure 5c).

These data clearly demonstrate that our murine lymphoma model fulfils many criteria of a classical DHL despite not being of GC origin (e.g. high levels of both MYC and BCL-2 expression, accelerated aggressive tumour onset, CD19 expression, monoclonal and enhanced activity of the cellular pathways found activated in the human DHL counterparts). We postulate that the translocations that lead to high BCL-2 and high MYC expression most likely arise in GC B cells and that, as evidenced by the RNA seq analysis, the high expression of these genes drives the tumour signature rather than the B cell of origin. Therefore, we believe the DHL-like lymphoma model we have generated will be very useful to test new therapeutic approaches *in vitro* and most importantly also *in vivo* that are of relevance for human DHL. Throughout the manuscript we have referred to our lymphomas as DHL-like to accurately represent them.

2. The authors state that they have made the first DHL model. Even if they can prove that their models are DHL, the statement is not true. Both humaneered and PDX models of DHL have been published by groups in Boston.

We thank the reviewer for this comment. Thus far we are the only group that has developed a mouse model for DHL, which enables us to grow the cells in culture and in syngeneic fully immune competent mice. This allows us to test standard chemotherapies, such as R-CHOP, alone or in combination with immune therapies to identify treatment regimens useful for clinical practice. To respond to this request from reviewer #1, we have incorporated additional text into the introduction to acknowledge these other models and we discuss the distinctions between these models and our new model.

3. The authors state that their unique lymphoma phenotype likely arises from lower expression of BCL2 compared to the Eu-Myc/Eu-Bcl2 models but they don't show any data to support it. The Eu-Myc/Eu-Bcl2 should be included as a comparator for the immunoblots (e.g. in Figures 1b, 3a, 4e, 5a).

This was a very valuable suggestion by the reviewer. We have performed this experiment and in fact could not see a clear difference in the levels of BCL-2 expression between E μ -Myc/E μ -BCL2 doubly transgenic lymphomas and our CRISPRa induced BCL-2 over-expressing lymphomas in the primary tumours. However, we need to keep in mind that BCL-2 in the Eu-Bcl-2 transgenic mouse model is of human origin whereas our model expresses the mouse BCL-2 protein. We selected an antibody that binds to both human BCL-2 and mouse BCL-2, but we cannot be certain that its affinity for human BCL-2 and mouse BCL-2 is the same.

One thing we did notice though, was that the BCL-2 protein levels in the cell lines derived from the E μ -Myc/dCas9a-SAM^{Kl/+}/sgBcl-2 lymphomas were lower than in the matched primary tumour cells. This capability to naturally modulate BCL-2 expression from the endogenous promoter may allow the derivation of cell lines which was not possible for the E μ -Myc/E μ -Bcl-2 lymphomas. Alternatively, the inability to derive cell lines from the lymphomas from the E μ -Myc/E μ -Bcl-2 doubly transgenic mice may relate to their immature progenitor type state. We have included the Western blot data demonstrating the aforementioned findings as Supplementary Fig. 8.

4. The replicates of genome-wide CRISPR-Cas9 screen are quite heterogeneous, as shown in Fig 6 and supplementary Fig 5. A statistical analysis should be included. The authors report enrichment of sgRNA upregulating of BCL-XL and MCL-1; however, the reads were only upregulated in very few replicates (supplementary Fig 5b). Further explanation is needed. Also, all of the CRISPR screen primary data, including all genes screened, should be included as a Supplementary Table.

In order to get low numbers of sgRNAs per cell and yet achieve maximal coverage of the library, we perform multiple independent infections of the lymphoma cells to be selected by treatment with venetoclax. Therefore, the number/identity of guides between 'replicates' in the pre-treatment group is actually quite diverse due to each replicate being transduced independently with only a subset of the guide RNA library. Therefore, we do not expect the same sgRNAs to be selected after venetoclax treatment in every individual replicate. This variability precludes us from performing a traditional statistical analysis (e.g. with Mageck). Therefore, we performed the statistical analysis described in the Methods section of our revised manuscript. Our analysis showed that sgRNAs targeting the genes for MCL-1 or BCL-XL were clearly present in many of the venetoclax resistant cells within a few of the replicates, however only sgRNAs targeting A1 were statistically significantly enriched in both cell lines when considering all replicate samples together. We now include in our revised manuscript an example of a box plot from this statistical analysis as Supplementary Fig. 11b, the full raw data set and a list of statistically significant genes from our screens are presented as Supplementary Data as requested by Reviewer #1.

5. The same is true for the MCL1 and BCL2 inhibitor data across cell lines. A supplementary table should be added with IC50 values so the reader can compare across lines.

We thank the reviewer for this suggestion and have now included these data in Supplementary Fig. 10b and d and Supplementary Table 1 of our revised manuscript.

Minor comments

1. In Fig. 1b, the control in lane 5 should be treated with puromycin instead of no puromycin treatment? And there is a bubble covering the data in lane 5 for BCL-2. The authors should provide a clear result.

We have repeated the experiment and the Western blot analysis. We have additionally treated cells transduced with a non-targeting control sgRNA with puromycin. The new data are shown in Fig. 1b and Supplementary Fig. 1b of our revised manuscript.

2. Fig. 2b lacks sample information/conditions.

We have replaced Fig. 2b with data from a long range (LR)PCR experiment to determine that the dCas9a construct has indeed been inserted into the Rosa26 locus. Sample information and conditions are now included in the figure legends and methods of our revised manuscript.

3. *The controls in Fig 3b-f are questionable. The authors used Cd4-sgRNA as a negative control for Cd19 sgRNA and vice versa. The authors should use non-targeting sgRNA as a control, as they did in Fig 3a. What's more, the authors need to provide the statistical information and present the data (as in Fig 3c) using a dot column for all of the experiments, including Fig 4b, Fig 4d and Fig 5b.*

As requested by Reviewer #1, we have included statistical information and changed the styles of data display in Figures 3b-3f, 4b, 4d and 5b.

With regards to the appropriate controls, we thank Reviewer #1 for this comment but would respectfully disagree on this point. The controls are fully appropriate and, we believe they are better than a non-targeting sgRNA. Having control cells in which an active dCas9a-SAM is inducing expression of a different gene provides reassurance that the activity of dCas9a-SAM in the presence of a gene targeting sgRNA in a cell does not lead to a global unspecific gene activation.

4. *There is no information in the figure legend or the methods on mouse sample collection methods, including mouse ages and other details.*

We thank Reviewer #1 for picking up this oversight and have now added this information into the Methods section of our revised manuscript and information about the mice is now presented in Supplementary Table 4 of our revised manuscript.

5. *The authors should provide both vendors and catalog numbers of the reagents used in the study, for example, in the 'Cell culture' section*

We have followed the advice of Reviewer #1 and now present this information in our revised manuscript.

Reviewer #2 (Remarks to the Author):

The paper presented by Deng and colleagues describes a novel mouse model which allows gene activation by Cas9 expression from the Rosa26 locus and applies the in vivo system to gain insights in aggressive B cell lymphoma pathogenesis. Overall, the presented data is convincing and the manuscript is well written. However, the following limitations apply:

We thank Reviewer #2 for their positive comments about our manuscript and have addressed the limitations indicated.

Major concerns:

The dCas9a-SAMKI mouse generated by Deng and colleagues shares similarities to the CRISPR-activator mouse model published recently by Hunt et al. (Hunt, C., Hartford, S.A., White, D. et al. Tissue-specific activation of gene expression by the Synergistic Activation

Mediator (SAM) CRISPRa system in mice. Nat Commun 12, 2770 (2021)). The authors might want to discuss the pros/cons of the different animal strains in their manuscript.

We thank Reviewer #2 for this suggestion and have included other CRISPRa models in the discussion of the revised version of our revised manuscript.

The dCas9a-SAMKI mouse will be of interest for researchers in various disciplines. Thus, the proper targeting of the Rosa26 locus is essential to avoid unwanted side-effects. From the results in Fig 2B it is not clear whether the correct integration of the targeting cassette exclusively in the Rosa26 locus is validated. Additional experiments (Southern blot?) or the detailed description of the PCR approach are needed.

We have followed this advice from Reviewer #2 and performed long range PCR and Southern Blot analysis. We now present the data from these experiments in our revised manuscript in Fig. 2b and Supplementary Fig. 2. The data shown confirm correct integration of the targeting cassette into the Rosa26 locus.

Several analyses demonstrating the efficacy of the modified CRISPR approach for gene activation in B and T cells are shown (see Fig 3). In the reported experiments cells isolated from homozygous transgenic mice (dCas9a-SAMKI/KI) were used. In the future, the novel dCas9a mouse might be used in combination with other transgenic mouse strains, in a plethora of them the Rosa26 locus is modified as well. The analysis of heterozygous transgenic mice (dCas9a-SAMKI/+) would be most interesting -> how much dCas9 is expressed from a single allele? Is the amount of dCas9a still sufficient for gene activation (of active and/or silenced genes)?

We thank Reviewer #2 for this valuable suggestion and have included the analysis of the CRISPRa system in primary B and T cells and MEFs derived from heterozygous dCas9a-SAM adult mice and E14.5 embryos, respectively. These experiments show that we can induce expression of a normally silenced gene in cells with two copies or even one copy of the dCas9a-SAM system. These new data are now presented in Supplementary Fig. 5 of our revised manuscript.

The B cell lymphomas arising in Emu-MYC, dCas9a-SAMKI/+, BCL2 mice are poorly characterized. As CD19 and B220 co-expression are not restricted to mature B cells, the origin of the tumors is still obscure. Detailed analyses of the tumor cohorts (morphology/histology; proliferation; flow cytometry of B cell differentiation markers, e.g. IgM, IgD, CD43, AA4.1, CD38, FAS; SHM analysis, etc.) are critical to judge the value of the model.

We thank Reviewer #2 for this suggestion and performed a more detailed analysis in the fully malignant lymphoma cells. Analysis of the established lymphoma cell lines revealed an immature (CD43+) mostly surface Ig positive phenotype of these cells. This led us to the conclusion that the cell of origin (COO) for our lymphomas is B cell committed but is not the germinal centre and we would therefore not expect any mutation of the IgV genes. In addition, we performed RNAseq analysis of our lymphomas and compared their RNA expression profile to a B-ALL RNAseq data set available at WEHI. We find clear

differences between the Eu-Myc/dCas9a/sgBCL-2 lymphomas and B-ALL lymphomas, including up-regulation of genes and pathways associated with DLBCL, including those perturbed with DHL translocations (Figure 6a). Importantly, in a gene set enrichment analysis, we identified enrichment of the 'SHIPP DLBCL VS FOLLICULAR LYMPHOMA UP' gene set, which includes the top 50 up-regulated markers distinguishing human DLBCL from follicular lymphomas that are also of B cell origin (Figure 6). Analysis of Ig heavy chain type sequence clearly established our lymphomas are monoclonal (Figure 5c).

These data demonstrate that our murine lymphoma model fulfils many criteria of a classical DHL despite not being of GC origin (e.g. high levels of both MYC and BCL-2 expression, accelerated aggressive tumour onset, CD19 expression, monoclonal and enhanced activity of the cellular pathways found activated in the human DHL counterparts). We postulate that the translocations that lead to high BCL-2 and high MYC expression most likely arise in GC B cells and that, as evidenced by the RNA seq analysis, the high expression of these genes drives the tumour signature rather than the B cell of origin. Therefore, we believe the DHL-like lymphoma model we have generated will be very useful to test new therapeutic approaches *in vitro* and most importantly also *in vivo* that are of relevance for human DHL. Where appropriate we have now referred to our lymphomas as DHL-like to accurately represent them.

In addition, human Double-hit-lymphomas (DHL) are an aggressive subgroup of DLBCL. Thus, antigen-activated B cells and the GC reaction are the origin of malignant transformation. Unfortunately, data about this cell population in dCas9a-SAMKI animals and in pre-malignant Emu-MYC, dCas9a-SAMKI/+, BCL2 mice are missing.

We followed the recommendation of Reviewer #2 and performed a detailed analysis of pre-leukaemic mice. We found many more B cells in mice transplanted with Eu-Myc/dCas9a/sgBCL-2 transduced HSPCs as compared to control mice. They were mostly of naïve origin, and we did not see any increase in germinal centre (GC) B cells. Hence, our conclusion is that the cells of origin of these lymphomas are not germinal centre B cells but most likely immature B lymphoid cells in the bone marrow. The RNA seq analysis we performed revealed that our DHL-like tumours expressed gene signatures consistent with human DLBCL/DHL, we therefore postulate that the high MYC/high BCL-2 expression that arises from translocations that may occur in the GC, are a more defining feature of DHL than the cell of origin. To accurately represent this, we refer to our lymphomas throughout the manuscript as DHL-like.

In DHL MYC and BCL2 are typically affected by genetic rearrangements which bring the oncogenes under the control of "foreign" promotor/regulator elements. Thus, the authors' selection of studying DHL pathogenesis for validating their dCas9 approach (in which gene expression from the endogenous context is achieved) is unclear.

We thank Reviewer #2 for this comment. We chose the DHL model as discussions with our clinical colleagues revealed that there is a desperate need to study this disease type and versatile murine models did not exist. In our model, MYC is under the control of a

foreign enhancer and BCL-2 is under the control of its own promoter, albeit still aberrantly over-expressed. Importantly, this leads to aggressive lymphomas that have many features of human DHL. In addition, human so called double expressor lymphomas (DEL) are clinically very similar to DHL, but in DELs, MYC and BCL-2 are not rearranged to a different promoter/enhancer but are aberrantly highly expressed from their own promoters. Hence, whether the genes are translocated versus overexpressed does not seem to affect the clinical features of the human disease. Importantly, mimicking the chromosomal rearrangements for MYC and BCL-2 in a murine model by putting both genes under the control of a foreign promoter/enhancer, namely the IgH enhancer ($E\mu$), has been done previously but this caused lymphomas of a progenitor phenotype that were not B cell committed and these lymphomas could not be grown in culture so were not amenable to *in vitro* studies (Strasser, Nature 1990). In conclusion, we believe that our new lymphomas provide a useful model of high MYC/high BCL-2 aggressive lymphomas that can help improve therapeutic strategies and outcomes for human DHL.

Minor concerns:

In B cells the dCas9a-based activation of already active genes is more efficient than in T cells (for silenced genes, opposite results are demonstrated). In the discussion (biological?) explanations for these findings can be discussed.

We thank Reviewer #2 for this comment and have now included some text in the discussion section of our revised manuscript.

Absolute numbers are missing in Fig S2, although this data is stated in the text (Page 6, line 166).

We have now added the absolute numbers from this experiment into Supplementary Fig.4 of our revised manuscript. Please note that the previous Supplementary Fig. 2 is now Supplementary Fig. 4.

Reviewer #3 (Remarks to the Author):

Deng and Diepstraten et al describe development of a new CRISPRa mouse which they utilize to create a new mouse cancer model of double hit MYC+/BCL2+ lymphomas and then subsequently to better understand tumor response to a BCL2 inhibitor, venetoclax. The authors show this CRISPRa mouse is useful and should be widely adopted. A big part of the impact of this paper will be through adoption of this mouse model. As such it is critically important for this mouse to be made available through Jackson labs or some other means. The authors would strengthen this manuscript by showing CRISPRa proteins are expressed and active in cell types or tissues beyond the hematopoietic system.

We thank Reviewer #3 for this suggestion, and we will certainly make our CRISPRa mice publicly available to the wider scientific community. In fact, we already had many enquiries for our model; so deposition of these new mice at the Jackson lab is a very good idea. Regarding expression of our system in other tissues than the blood, we have performed Western blot analysis for Cas9 in diverse organs including liver, kidney and

heart. This revealed its uniform expression and additional tests demonstrated efficient gene induction in MEFs. These data are presented in Supplementary Fig. 4 and 5 of our revised manuscript.

The authors should discuss why this new SAM CRISPRa mouse is different or better than other CRISPRa mice such as PMID: 29335603, 30545847 or preprint: doi: <https://doi.org/10.1101/2021.03.08.434430>. One way to discuss this could be in the context of: PMID: 27214048.

We thank Reviewer #3 for this suggestion, and we have now included a more detailed discussion of these previously reported strains of mice into the discussion of our revised manuscript.

In the introduction the authors discuss shortcomings of previous double hit lymphoma mouse models. The authors should consider citing efforts to create humanized models of double hit lymphomas which do express B cell markers and have been useful for understanding response to anti-cancer therapies (PMID: 22484426 and 24485462). The authors claim that the Eumyc/sgBCL2 tumors are a resource for understanding double hit lymphoma biology and drug response but they need to justify why this model is better than the human double hit model above. One justification could relate to the fact you can use the Eumyc/sgBCL2 model in a fully immunocompetent syngeneic mouse which is not possible with the humanized model.

We agree with Reviewer #3. We have now included a description of the humanised mouse models of DHL and we discuss the advantages of our new mouse model in terms of being transplantable into immunocompetent mice in the introduction and discussion sections of our revised manuscript.

The authors state that the SAM mouse does not have developmental issues or overt disease which is great but this analysis could be more nuanced. For example, recently analysis of a dCas9-p300 mouse (<https://doi.org/10.1101/2021.03.08.434430>) has revealed off target activity when looking at RNA-seq or CHIP-seq data comparing saline injected mice to sgRNA expressing mice. Do you see any transcriptional differences +/- sgRNA expression? The comparison of saline to a negative control sgRNA via RNA-seq would illuminate possible off target activity. This is important as this mouse also represents a potential resource for others in the field and so it is important to define upfront potential strengths and problems.

We have followed the advice of Reviewer #3 and have compared the gene expression pattern by RNAseq of MEFs derived from wildtype mice with MEFs from our dCas9-SAM^{KI/+} mice, both on a C57BL/6 background. These data are now presented in Supp Figure 3d of our revised manuscript. No major differences in gene expression were detected between WT MEFs and dCas9-SAM^{KI/+} transduced with a non-targeting sgRNA.

Related to Figure 4,5- do the Eu-myc/sgBCL2 lymphomas develop additional secondary mutations in the TP53 pathways as is frequently the case in the Eumyc model? I ask because

the comparison to nontargeting control lymphomas will often be a comparison to a lymphoma that has lost TP53,Arf or overexpressed MDM2 according to the older literature.

We followed the suggestion of Reviewer #3 and analysed 10 NTC and 10 sgBCL-2 sgRNA lymphomas for TP53 aberrations. This showed no difference in the percentages of lymphomas carrying mutations in *Trp53*. This is in accordance with human DHL samples in which TP53 mutations were identified (Gebauer N, Leuk Lymphoma 2015). The new data are now included in Supplementary Fig. 7b of our revised manuscript.

The observation that BFL-1 upregulation promotes resistance to venetoclax is really interesting as this gene is understudied. It would greatly strengthen this observation if there is patient data for double hit lymphomas treated with venetoclax to support this observation.

We agree that this is a very exciting discovery and we thank Reviewer #3 for acknowledging this. Based on discussions with clinical colleagues, we believe there is anecdotal evidence of BFL-1 upregulation in venetoclax resistant patient samples in AML and CLL and as such many pharmaceutical companies are starting to invest in the development of BFL-1 targeting BH3 mimetic drugs. However, acquired resistance to venetoclax is really only just emerging in patients; so such samples are limited and we are therefore unable to include such data here.

REVIEWERS' COMMENTS

Reviewer #1 (Remarks to the Author):

The authors have responded very completely to my concerns.

Reviewer #2 (Remarks to the Author):

My comments were addressed sufficiently.

Thank you!

Reviewer #3 (Remarks to the Author):

Deng and Diepstraten et al's response to review addresses most of my concerns. The authors state that it would be "a good idea" to deposit this mouse on Jackson but do not commit to this. I would strongly urge the authors to commit to depositing this mouse in a repository to maximize the utility of this as a resource for the community of folks who are likely to be interested in the mouse.

The RNA-seq data in Supplementary Figure 3D is a valuable addition however no details on the analysis methods are given in the figure legend or the methods. Additional details are required to ensure this analysis is robust.

For ease of reading, the comments from the reviewers are presented in black italics; and our responses are presented in plain blue script.

REVIEWER COMMENTS

Reviewer #1 (Remarks to the Author):

The authors have responded very completely to my concerns.

We thank Reviewer #1 for their positive comments about our revised manuscript.

Reviewer #2 (Remarks to the Author):

My comments were addressed sufficiently.
Thank you!

We thank Reviewer #2 for their positive comments about our revised manuscript.

Reviewer #3 (Remarks to the Author):

Deng and Diepstraten et al's response to review addresses most of my concerns. The authors state that it would be "a good idea" to deposit this mouse on Jackson but do not commit to this. I would strongly urge the authors to commit to depositing this mouse in a repository to maximize the utility of this as a resource for the community of folks who are likely to be interested in the mouse.

We thank Reviewer #3 for their comments. We will initiate the discussions on depositing our model to the Jackson lab for wider distribution of our mouse model.

The RNA-seq data in Supplementary Figure 3D is a valuable addition however no details on the analysis methods are given in the figure legend or the methods. Additional details are required to ensure this analysis is robust.

We thank the reviewer for their suggestion and have added a statement in the Methods section under statistical analysis to clarify the analysis pipeline presented of how Supplementary Figure 3D was analysed. In addition we have made all our RNAseq data and codes available via GEO and Github, respectively, which we have also included in the revised version of our manuscript.